# PTEN controls glandular morphogenesis through a juxtamembrane β-Arrestin1/ARHGAP21 scaffolding complex

**Arman Javadi[1†], Ravi K Deevi[1†], Emma Evergren[1], Elodie Blondel-Tepaz[2,3,4], George S Baillie[5], Mark GH Scott[2,3,4], Frederick C Campbell[1]***

[1]Centre for Cancer Research and Cell Biology, Queen's University of Belfast, Belfast, United Kingdom; [2]Inserm, U1016, Institut Cochin, Paris, France; [3]CNRS, UMR8104, Paris, France; [4]Univ. Paris Descartes, Sorbonne Paris Cité, Paris, France; [5]Institute of Cardiovascular and Medical Science, College of Medical, Veterinary and Life Sciences, University of Glasgow, Glasgow, Scotland

**Abstract** PTEN controls three-dimensional (3D) glandular morphogenesis by coupling juxtamembrane signaling to mitotic spindle machinery. While molecular mechanisms remain unclear, PTEN interacts through its C2 membrane-binding domain with the scaffold protein β-Arrestin1. Because β-Arrestin1 binds and suppresses the Cdc42 GTPase-activating protein ARHGAP21, we hypothesize that PTEN controls Cdc42 -dependent morphogenic processes through a β-Arrestin1-ARHGAP21 complex. Here, we show that PTEN knockdown (KD) impairs β-Arrestin1 membrane localization, β-Arrestin1-ARHGAP21 interactions, Cdc42 activation, mitotic spindle orientation and 3D glandular morphogenesis. Effects of PTEN deficiency were phenocopied by β-Arrestin1 KD or inhibition of β-Arrestin1-ARHGAP21 interactions. Conversely, silencing of ARHGAP21 enhanced Cdc42 activation and rescued aberrant morphogenic processes of PTEN-deficient cultures. Expression of the PTEN C2 domain mimicked effects of full-length PTEN but a membrane-binding defective mutant of the C2 domain abrogated these properties. Our results show that PTEN controls multicellular assembly through a membrane-associated regulatory protein complex composed of β-Arrestin1, ARHGAP21 and Cdc42.

DOI: https://doi.org/10.7554/eLife.24578.001

**\*For correspondence:**
f.c.campbell@qub.ac.uk

[†]These authors contributed equally to this work

**Competing interests:** The authors declare that no competing interests exist.

## Introduction

*PTEN* (phosphatase and tensin homolog) is the second most commonly mutated tumor suppressor gene in human cancer (*Cantley and Neel, 1999*) and has a central role in multicellular morphogenesis (*Martin-Belmonte et al., 2007*; *Jagan et al., 2013a*; *Deevi et al., 2016*). While PTEN antagonizes the phosphoinositol 3-kinase (PI3K)/AKT pathway via its N-terminal phosphatase domain (*Cantley and Neel, 1999*), three-dimensional (3D) multicellular assembly was unaffected by forced variation of PI3K activity in colorectal organotypic model systems (*Jagan et al., 2013a*; *Magudia et al., 2012*). The *PTEN* domain structure includes an N-terminal phosphatase domain, a C2 domain, a C-terminal tail and a PDZ-binding domain. The C2 domain binds to membrane phospholipids by inserting a hydrophobic (CBR3) loop into the membrane bilayer and thereby provides a scaffold for juxtamembrane signaling (*Lee et al., 1999*). Furthermore, the PTEN C2 domain regulates polarized migration (*Raftopoulou and Hall, 2004*), multicellular morphology (*Leslie et al., 2007*; *Jagan et al., 2013b*) and has an important but poorly understood tumor suppressor function (*Caserta et al., 2015*).

Within complex systems, protein scaffolding enhances signaling efficiency by assembly of spatially distinct subcellular complexes for different cellular tasks (*Weng et al., 1999*; *Pertz, 2010*). The

**eLife digest** The protein PTEN helps to organize cells in the body to form complex structures. In particular, it collects signals from a cells' surroundings and changes where cells divide so new cells are produced in the right places. The control of cell division by PTEN is also thought to help limit the progression and spread of cancer.

PTEN can interact with another protein called β-Arrestin1, which behaves as a so-called scaffolding protein – in other words, one that helps groups of proteins to interact with each other. β-Arrestin1 has been found to control cell division via a series of other proteins, including ARHGAP21 and Cdc42. The relationship between PTEN and these other proteins in dividing cells is still not fully understood.

Javadi, Deevi et al. studied PTEN in human cells grown in the laboratory to show that a part of PTEN known as the C2 domain allows it to help organize cells by moving β-Arrestin1 to the outer edge of the cell – the cell membrane. This relocation allows β-Arrestin1 to interact with ARHGAP21 and Cdc42, and control cell division. Active Cdc42 changes the orientation of cell division, allowing cells to organize into single layers of regular cells and similar tightly controlled structures.

Further experiments revealed that these proteins are important to form tubes inside the glands of the gut. The C2 region of PTEN also helps to detect signals carried by fat molecules in the cell membrane, so these results provide a direct link between signaling and cell organization via PTEN. The work of Javadi, Deevi et al. provides new understanding of how PTEN links nutrient availability to cell organization during development and may also lead to new insights into the role of PTEN in limiting the growth of tumors.

DOI: https://doi.org/10.7554/eLife.24578.002

PTEN C2 domain binds the plasma membrane and interacts with the scaffold protein β-Arrestin1 (*Lima-Fernandes et al., 2011*) that in turn binds and suppresses ARHGAP21 (*Anthony et al., 2011*), a member of a highly conserved class of RhoGAPs (*Bos et al., 2007*; *Anderson et al., 2008*). ARHGAP21 regulates the small GTPases, Cdc42 (*Dubois et al., 2005*) and RhoA (*Anthony et al., 2011*). These GTPases have overlapping, complementary functions required for mitotic spindle orientation and consequent control of the cell division axis, cytokinetic furrow positioning, daughter cell size and tissue morphogenesis (*Morin and Bellaïche, 2011*). Both Cdc42 and RhoA drive actin nucleation and cortical stiffening (*Ma et al., 1998*; *Eisenmann et al., 2007*) required for spindle orientation (*Johnston et al., 2013*). Furthermore, Cdc42 crosstalk with protein kinase c zeta [PRKCZ] (*Noda et al., 2001*; *Durgan et al., 2011*) localizes force generators within the cell cortex that act via astral microtubules to orientate the spindle (*Hao et al., 2010*). ARHGAP21 has high GAP activity for Cdc42 (*Dubois et al., 2005*) and its Pac-1 homologue regulates multicellular patterning in C. elegans by spatial regulation of Cdc42 (*Anderson et al., 2008*; *Klompstra et al., 2015*).

Here, we investigate PTEN spatiotemporal coordination of mammalian glandular morphogenesis through conserved juxtamembrane β-Arrestin1-ARHGAP21 interactions, using 3D colorectal cancer (CRC) model systems. To substantiate physiological relevance of these processes, we also investigate their role in morphogenesis of 3D multicellular organoids isolated from normal colon.

## Results

### PTEN regulates β-Arrestin1 membrane localization

β-Arrestin1 scaffolds juxtamembrane signaling networks (*Kovacs et al., 2009*), binds ARHGAP21 (*Anthony et al., 2011*) and governs PTEN catalytic and noncatalytic functions (*Lima-Fernandes et al., 2011*). To ascertain whether PTEN regulates membrane-associated β-Arrestin1 and ARHGAP21, we conducted expression and simple fractionation studies in PTEN-expressing [Caco-2 and HCT116] and -deficient [Caco-2 Sh*PTEN* (Sh*PTEN*) and *PTEN* $^{-/-}$ HCT116 (*PTEN* $^{-/-}$)] cells. We found near-significant or significant differences of total lysate β-Arrestin1 and ARHGAP21 between PTEN-expressing and -deficient cells [Caco-2 vs Sh*PTEN* (*Figure 1A,B*) and HCT116 vs *PTEN* $^{-/-}$ cells (*Figure 1—figure supplements 1* and *2*)]. To infer subcellular localization of β-Arrestin1 and ARHGAP21, we performed membrane fractionation studies and normalized each

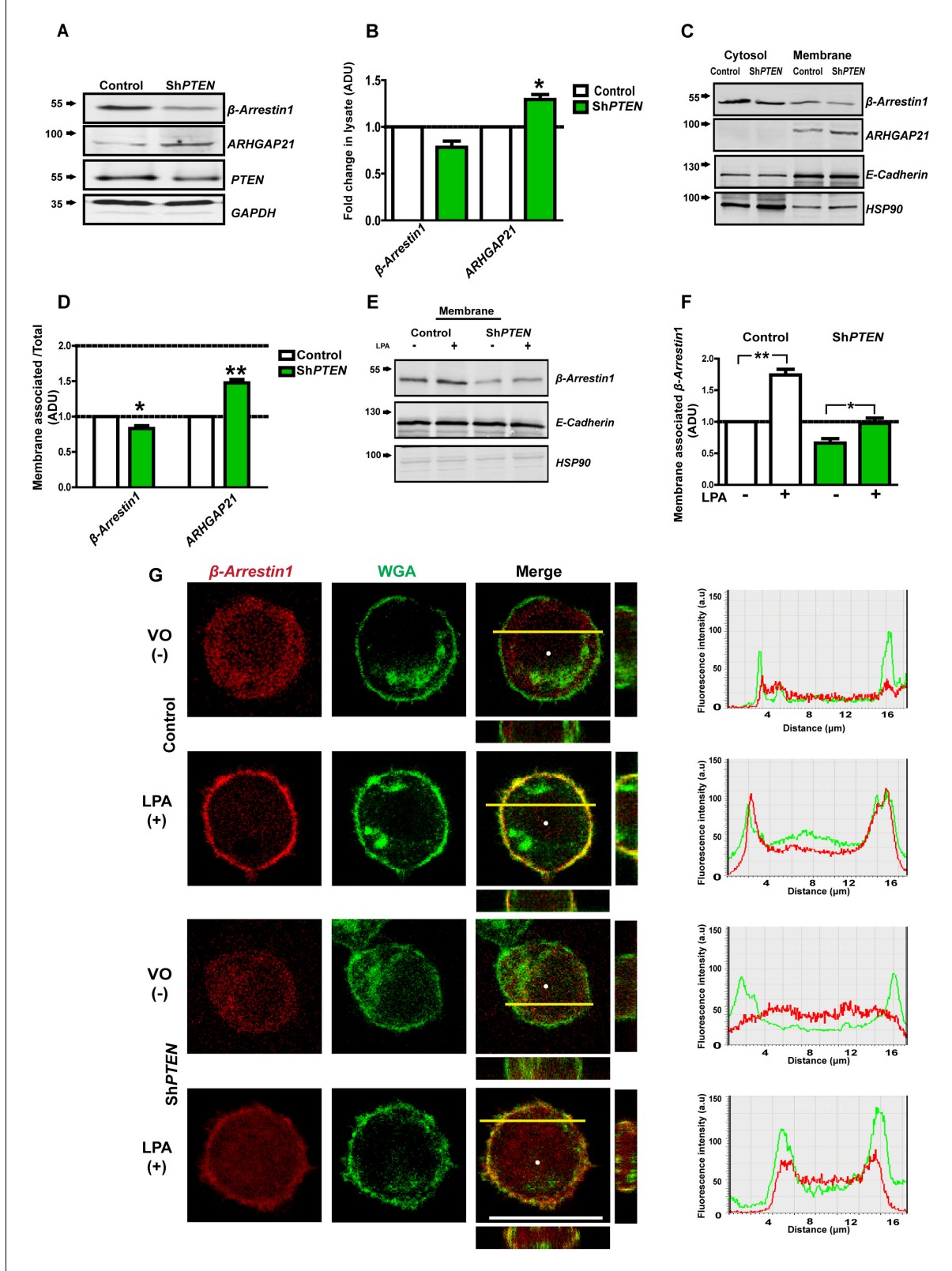

**Figure 1.** PTEN enhances membrane recruitment of β-Arrestin1. (**A,B**) Total lysate β-Arrestin1 and ARHGAP21 expression in ShPTEN vs Caco-2 (control). GAPDH loading control; β-Arrestin1 = 0.78 ± 0.06; p=NS (0.08); ARHGAP21 = 1.29 ± 0.05;*p=0.03. (**C**) Cytosol and membrane localization of β-Arrestin1 and ARHGAP21 in control Caco-2 or Caco-2 ShPTEN (ShPTEN) cells. E-Cadherin and HSP90 used as cell membrane or cytosol markers. (**D**) Summary β-Arrestin1 and ARHGAP21 membrane ADU in ShPTEN cells vs Caco-2 control. Membrane values are normalized against total lysate ADU

*Figure 1 continued on next page*

Figure 1 continued

values for each protein (β-Arrestin1 = 0.82 ± 0.03 *p<0.05; ARHGAP21 = 1.48 ± 0.05 **p<0.01). (E) β-Arrestin1 expression in membrane fractions of control Caco-2 and ShPTEN cells after treatment. (F) Summary fold change of membrane β-Arrestin1 after LPA (+) or VO [vehicle only(-)] treatment shown in (E) ;Caco-2 (+) vs (-)=1.74 ± 0.09; **p<0.01; ShPTEN (+) vs ShPTEN (-)=0.97 ± 0.08 vs 0.66 ± 0.07;*p<0.05 values expressed as fold change relative to Caco-2 (-) control. ShPTEN (-) vs Caco-2 (-)=0.66 ± 0.07;*p<0.05; ShPTEN (+) vs Caco-2 (+)=0.97 ± 0.08 vs 1.74 ± 0.09;**p<0.01. Green bars indicate ShPTEN cells. (G) Plasma membrane localization of the β-Arrestin1-mCherry fusion protein in Caco-2 control (top two panels) or ShPTEN (bottom two panels) after VO (-) or LPA (+) treatment. Red and green fluorescence emitted by m-Cherry and Alexa 488 labels correspond to β-Arrestin1 and WGA, respectively. Note WGA localization to plasma membranes and Golgi apparatus. Colocalization of β-Arrestin1 and WGA at the plasma membrane revealed by composite yellow signal in Merge, Z-stack images and by overlap of fluorescence intensity curves in line scans. Horizontal yellow bars indicate focal plane. All experiments in triplicate. Analyses by Student's paired t test or ANOVA with Tukey post hoc test. White scale bar 20 μm. Molecular weights indicated by arrows in blots.

DOI: https://doi.org/10.7554/eLife.24578.003

The following source data and figure supplements are available for figure 1:

**Source data 1.** Source data for *Figure 1B*.
DOI: https://doi.org/10.7554/eLife.24578.010
**Source data 2.** Source data for *Figure 1D*.
DOI: https://doi.org/10.7554/eLife.24578.011
**Source data 3.** Source data for *Figure 1F*.
DOI: https://doi.org/10.7554/eLife.24578.012
**Figure supplement 1.** Total β-Arrestin1 and ARHGAP21 expression in control HCT116 and *PTEN* $^{-/-}$ cell lysates.
DOI: https://doi.org/10.7554/eLife.24578.004
**Figure supplement 2.** Cell lysate expression of β-Arrestin1 and ARHGAP21 shown in *Figure 1—figure supplement 1*.
DOI: https://doi.org/10.7554/eLife.24578.005
**Figure supplement 2—source data 1.** . *Figure 1—figure supplement 2* Beta-Arestin1 and ARHGAP21 expression in HCT116 clones -Source data.
DOI: https://doi.org/10.7554/eLife.24578.013
**Figure supplement 3.** Treatment effects on membrane β-Arrestin in HCT116 control vs *PTEN* $^{-/-}$ cells.
DOI: https://doi.org/10.7554/eLife.24578.006
**Figure supplement 4.** Treatment effects on membrane β-Arrestin1 shown in *Figure 1—figure supplement 3*.
DOI: https://doi.org/10.7554/eLife.24578.007
**Figure supplement 4—source data 1.** *Figure 1—figure supplement 4* LPA effects on Beta-Arrestin1 in HCT116 clones.
DOI: https://doi.org/10.7554/eLife.24578.014
**Figure supplement 5.** Plasma membrane localization of the β-Arrestin1-mCherry fusion protein in HCT116 control (top two panels) or *PTEN* $^{-/-}$ cells (bottom two panels) after VO (-) or LPA (+) treatment.
DOI: https://doi.org/10.7554/eLife.24578.008
**Figure supplement 6.** Control mCherry distribution in HCT116 and *PTEN* $^{-/-}$ cells.
DOI: https://doi.org/10.7554/eLife.24578.009

protein's densitometry value against its total lysate level, to investigate relative proportions of β-Arrestin1 and ARHGAP21 associated with membrane. We found greater β-Arrestin1 but lower ARHGAP21 levels in Caco-2 than in Sh*PTEN* membrane fractions (*Figure 1C,D*). As β-Arrestins are known to localize to activated lysophosphatidic acid receptors [LPARs] (*Urs et al., 2005*; *Li et al., 2009*) that are expressed in Caco-2 and HCT116 cell membranes (*Yun et al., 2005*), we investigated effects of PTEN on lysophosphatidic acid (LPA)-induced membrane recruitment of β-Arrestin1. We found greater LPA-mediated membrane enrichment of β-Arrestin1 in Caco-2 and HCT116 cells than in PTEN-deficient Sh*PTEN* or *PTEN* $^{-/-}$ HCT116 (*PTEN* $^{-/-}$) subclones (*Figure 1E,F*; *Figure 1—figure supplements 3* and *4*). We next used confocal microscopy to determine PTEN effects on β-Arrestin1 subcellular distribution in whole cells. We expressed the β-Arrestin1-mCherry fusion protein and mCherry only controls in PTEN-expressing and -deficient cells. We assessed colocalization with Alexa 488-labeled wheat germ agglutinin (WGA), a widely used fluorescent probe for cell and Golgi complex membranes (*Crossman et al., 2015*) by confocal microscopy. β-Arrestin1-mCherry was predominantly cytosolic in vehicle only (VO)-treated cells, in accord with cytosolic accumulation of unlabelled β-Arrestin1 in fractionation studies. On treatment with LPA, the β-Arrestin1-mCherry fusion protein colocalized with WGA at the plasma membrane in PTEN-expressing Caco-2 and HCT116 control cells (*Figure 1G*; *Figure 1—figure supplement 5*). Line scanning analysis revealed overlap of β-Arrestin1-mCherry and Alexa 488 fluorescence signals in plasma membrane peaks in PTEN-expressing Caco-2 and HCT116 cells after LPA treatment (*Figure 1G*; *Figure 1—figure supplement 5*).

While LPA had limited effects in Sh*PTEN* cells that have residual low level PTEN (*Figure 1G*), this treatment had no effects on β-Arrestin1-mCherry subcellular distribution in *PTEN*-null (*PTEN* [-/-]) cells (*Figure 1—figure supplement 5*). mCherry only did not localize at the plasma membrane (data shown for control PTEN-expressing HCT116 and *PTEN* [-/-] cells only; *Figure 1—figure supplement 6*). To exclude a non-specific effect of PTEN on ligand-mediated protein translocation to the cell membrane, we investigated 1,25$(OH)_2D_3$-mediated membrane localization of E-Cadherin (*Pálmer et al., 2001*) in Caco-2 and Sh*PTEN* cells. We found that 1,25$(OH)_2D_3$ treatment induced equivalent E-Cadherin translocation to the plasma membrane in PTEN-expressing and -deficient cells, compared to control VO treatment (data not shown). Collectively, these findings show that PTEN functions within a regulatory scaffolding network that couples β-Arrestin1 to ARHGAP21 at the plasma membrane.

## PTEN controls Cdc42-dependent epithelial morphogenesis through β-Arrestin1-ARHGAP21 interactions

Within signaling scaffolds, β-Arrestin1 regulates monomeric GTPases (*Barnes et al., 2005*) and orchestrates cytoskeletal rearrangements (*Ge et al., 2003*). We investigated β-Arrestin1-ARHGAP21 coregulation of Cdc42, mitotic spindle orientation and morphogenesis in 3D organotypic model systems. SiRNA knockdown (KD) of β-Arrestin1 in control PTEN-expressing Caco-2 cells suppressed Cdc42 activation as assessed by Cdc42-GTP levels in cell lysates on Western blots (*Figure 2A,B*). In contrast, siRNA KD of ARHGAP21 enhanced Cdc42 activation in PTEN-deficient cells (*Figure 2C,D*). During normal organotypic 3D glandular morphogenesis, mitotic spindle planes are orientated at approximately right angles to gland centres (GCs) by Cdc42-dependent mechanisms. Conversely, Sh*PTEN* cells show deficiencies of these processes (*Jagan et al., 2013a*; *Deevi et al., 2016*; *Jagan et al., 2013b*). In 3D Caco-2 cultures, SiRNA β-Arrestin1 KD suppressed *Cdc42*-GTP (*Figure 2E,F*), induced mitotic spindle misorientation and abnormal multilumen formation (*Figure 2E*, *Figure 2—figure supplements 1* and *2*). Conversely, ARHGAP21 KD enhanced Cdc42-GTP (*Figure 2G,H*), restored mitotic spindle orientation and promoted single lumen formation in 3D Sh*PTEN* cultures (*Figure 2G*, *Figure 2—figure supplements 3* and *4*). Because of the previously reported relationship between ARHGAP21 and RhoA (*Lima-Fernandes et al., 2011*), we assessed relationships between β-Arrestin1, ARHGAP21 and RhoA. We found that activation of RhoA related directly to β-Arrestin1 and inversely to ARHGAP21 expression. β-Arrestin1 and RhoA-GTP were suppressed, while ARHGAP21 expression was enhanced by PTEN knockdown (data for RhoA-GTP not shown). Taken together, these data indicate that PTEN regulates β-Arrestin1-ARHGAP21 interactions to control GTPase signaling, mitotic spindle orientation and 3D multicellular morphology.

## PTEN promotes β-Arrestin1 and ARHGAP21 interactions through its C2 domain

β-Arrestin1 has previously been shown to bind the PTEN C2 domain directly and modulate PTEN function (*Lima-Fernandes et al., 2011*). To investigate PTEN regulation of β-Arrestin1-ARHGAP21 interactions, we conducted co-immunoprecipitation (CoIP) studies and normalized β-Arrestin1-associated ARHGAP21 against total ARHGAP21 densitometry values in cell lysates. Here, we show greater β-Arrestin1-associated ARHGAP21 levels in PTEN-expressing Caco-2 or HCT116 cells versus Sh*PTEN* or *PTEN*[-/-] cells or IgG negative controls (*Figure 3A,B*). To investigate involvement of *PTEN* catalytic and noncatalytic domains in these processes, we conducted transient expression studies of GFP-labeled full-length wild type (wt) *PTEN* or mutants (*Figure 3C*), in *PTEN*-deficient cells. Mutants included full-length *PTEN* with a mutation at the CBR3 membrane-binding loop within the C2 domain (*PTEN*-MCBR3), full-length phosphatase-dead (*PTEN* C124S-based) constructs with mutations in key C-terminal phosphorylation sites, namely *PTEN* C124S-T383A (CS-T383A) and *PTEN* C124S-A4 (CS-A4 with S380A, T382A, T383A and S385A mutations combined). CS-T383A has been proposed to contain an unmasked C2 domain (*Raftopoulou et al., 2004*) that effectively binds β-Arrestin1 while CS-A4 lacks β-Arrestin1 binding capacity (*Lima-Fernandes et al., 2011*). We also used the isolated *PTEN* C2 domain (C2) and a membrane-binding mutant of the C2 domain (C2-MCBR3). We found that expression of C2 enhanced β-Arrestin1-associated ARHGAP21 levels in CoIPs conducted in Sh*PTEN* (*Figure 3D,E,*) and *PTEN* [-/-] cells (*Figure 3—figure supplements 1* and *2*). Conversely, C2-MCBR3 had no significant effect on β-Arrestin1-associated ARHGAP21 levels in

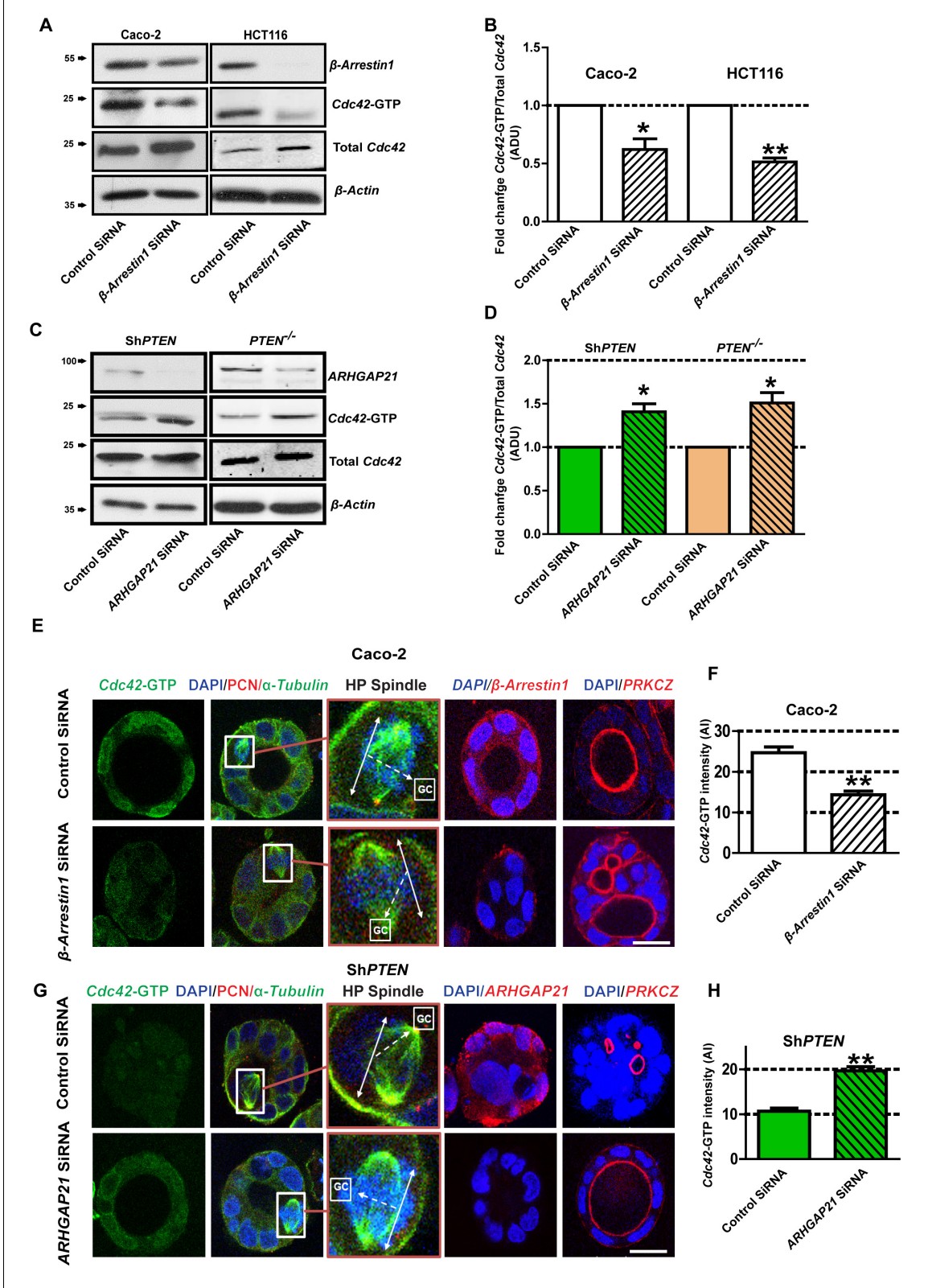

**Figure 2.** PTEN controls morphogenesis through Cdc42/β-Arrestin1/ARHGAP21 interactions. (**A,B**) SiRNA β-Arrestin1 knockdown (KD) suppresses Cdc42-GTP in Caco-2 and HCT116 cells (fold changes = 0.62 ± 0.09;*p=0.03 and 0.51 ± 0.03;**p=0.005 respectively. (**C,D**) SiRNA ARHGAP21 KD enhances Cdc42-GTP in Caco-2 Sh*PTEN* (Sh*PTEN*) and HCT116 *PTEN*⁻/⁻ (*PTEN*⁻/⁻) cells (fold change = 1.41 ± 0.09 and 1.51 ± 0.11, respectively; *p=0.02 for each. Cdc42-GTP ADU was normalized against total Cdc42. (**E**) SiRNA β-Arrestin1 KD suppresses Cdc42-GTP signal intensity, impairs spindle

*Figure 2 continued on next page*

*Figure 2 continued*

orientation and inhibits single lumen formation. High-power (HP) spindle views (orange border) enlarge areas within white rectangles and show orientation angles (interrupted white arrows) of spindle planes (double-headed solid white arrows) toward gland centres (GCs). Normal spindle planes are orientated at approximately $90^0$ angles relative to gland centres [GCs] (*Jaffe et al., 2008*). Summary SiRNA effects on spindle angles relative to GCs are shown in *Figure 2—figure supplement 1*. (F) Summary SiRNA effects on Cdc42-GTP intensity shown in (E) - control vs *β-Arrestin1* SiRNA = 24.67 ± 1.45 vs 14.33 ± 0.88 AI units; **p=0.004. *β-Arrestin1* KD also suppresses single central lumen formation in 3D Caco-2 cultures (E; *Figure 2—figure supplement 2*). (G) SiRNA *ARHGAP21* KD increases Cdc42-GTP signal intensity (H), rescues spindle orientation (G, *Figure 2—figure supplement 3*) and central lumen formation in Sh*PTEN* 3D cultures (G, *Figure 2—figure supplement 4*). (H) Cdc42-GTP, control vs SiRNA *ARHGAP21* KD in Sh*PTEN* cultures = 10.67 ± 0.67 vs=19.67 ± 0.88 AI units; **p<0.01. Assays at 4 days of culture. Imaging Cdc42-GTP [green], pericentrin (*PCN*) [red], α-Tubulin [green], ARHGAP21 [red], PRKCZ [red], β-Arrestin1 [red] and DAPI [blue]. All experiments conducted in triplicate. All analyses by paired Student's t test. Scale bars 20 μm. Molecular weights indicated by arrows in blots.

DOI: https://doi.org/10.7554/eLife.24578.015

The following source data and figure supplements are available for figure 2:

**Source data 1.** Source data for *Figure 2B*.
DOI: https://doi.org/10.7554/eLife.24578.020
**Source data 2.** Source data for *Figure 2D*.
DOI: https://doi.org/10.7554/eLife.24578.021
**Source data 3.** Source data for *Figure 2F*.
DOI: https://doi.org/10.7554/eLife.24578.022
**Source data 4.** Source data for *Figure 2H*.
DOI: https://doi.org/10.7554/eLife.24578.023
**Figure supplement 1.** Summary effects of siRNA *β-Arrestin1* or *ARHGAP21* KD vs control non-targeting SiRNA on mitotic spindle angles and lumen formation in 3D Caco-2 and Caco-2 Sh*PTEN* (Sh*PTEN*) cultures.
DOI: https://doi.org/10.7554/eLife.24578.016
**Figure supplement 1—source data 1.** *Figure 2—figure supplement 1* Spindle angles in Caco-2 after Beta-Arrestin1 KD.
DOI: https://doi.org/10.7554/eLife.24578.024
**Figure supplement 2.** Summary effects of siRNA *β-Arrestin1* or *ARHGAP21* KD vs control non-targeting SiRNA on mitotic spindle angles and lumen formation in 3D Caco-2 and Caco-2 Sh*PTEN* (Sh*PTEN*) cultures.
DOI: https://doi.org/10.7554/eLife.24578.017
**Figure supplement 2—source data 1.** *Figure 2—figure supplement 2* - Single central lumen fomation on Caco-2 after Beta-Arrestin1 KD.
DOI: https://doi.org/10.7554/eLife.24578.025
**Figure supplement 3.** Summary effects of siRNA *β-Arrestin1* or *ARHGAP21* KD vs control non-targeting SiRNA on mitotic spindle angles and lumen formation in 3D Caco-2 and Caco-2 Sh*PTEN* (Sh*PTEN*) cultures.
DOI: https://doi.org/10.7554/eLife.24578.018
**Figure supplement 3—source data 1.** *Figure 2—figure supplement 3* Spindle angles in ShPTEN after ARHGAP21 KD.
DOI: https://doi.org/10.7554/eLife.24578.026
**Figure supplement 4.** Summary effects of siRNA *β-Arrestin1* or *ARHGAP21* KD vs control non-targeting SiRNA on mitotic spindle angles and lumen formation in 3D Caco-2 and Caco-2 Sh*PTEN* (Sh*PTEN*) cultures.
DOI: https://doi.org/10.7554/eLife.24578.019
**Figure supplement 4—source data 1.** *Figure 2—figure supplement 4* Single central lumen in ShPTEN after ARHGAP21kd.
DOI: https://doi.org/10.7554/eLife.24578.027

CoIPs (*Figure 3D,E*, *Figure 3—figure supplements 1* and *2*). β-Arrestin1-associated ARHGAP21 levels were normalized against total *ARHGAP21* expression in cell lysates. Collectively, these findings indicate that the membrane-binding function of PTEN is important for scaffolding ARHGAP21 and β-Arrestin1.

We then used an intramolecular bioluminescence resonance energy transfer (BRET)-based PTEN biosensor (*Lima-Fernandes et al., 2014*; *Misticone et al., 2016*) to test if the full-length C124S C-terminal phosphorylation mutants (*Figure 3C*) that have different β-Arrestin1 binding capacities (*Lima-Fernandes et al., 2011*), display different conformations. The biosensor contains PTEN sandwiched between the energy donor Renilla luciferase (Rluc) and the energy acceptor YFP. Changes in the BRET signal depend on the relative distance and orientation of the donor and acceptor proteins within the fusion and therefore provide readout for conformational change of PTEN in live cells (*Figure 3—figure supplement 3*). Wild-type (wt) *PTEN*, CS-T383A and CS-A4 mutants in the Rluc-*PTEN*-YFP construct produced different BRET signals (*Figure 3—figure supplement 3*). These findings show that the phosphatase-dead mutants do indeed adopt different conformations, which is

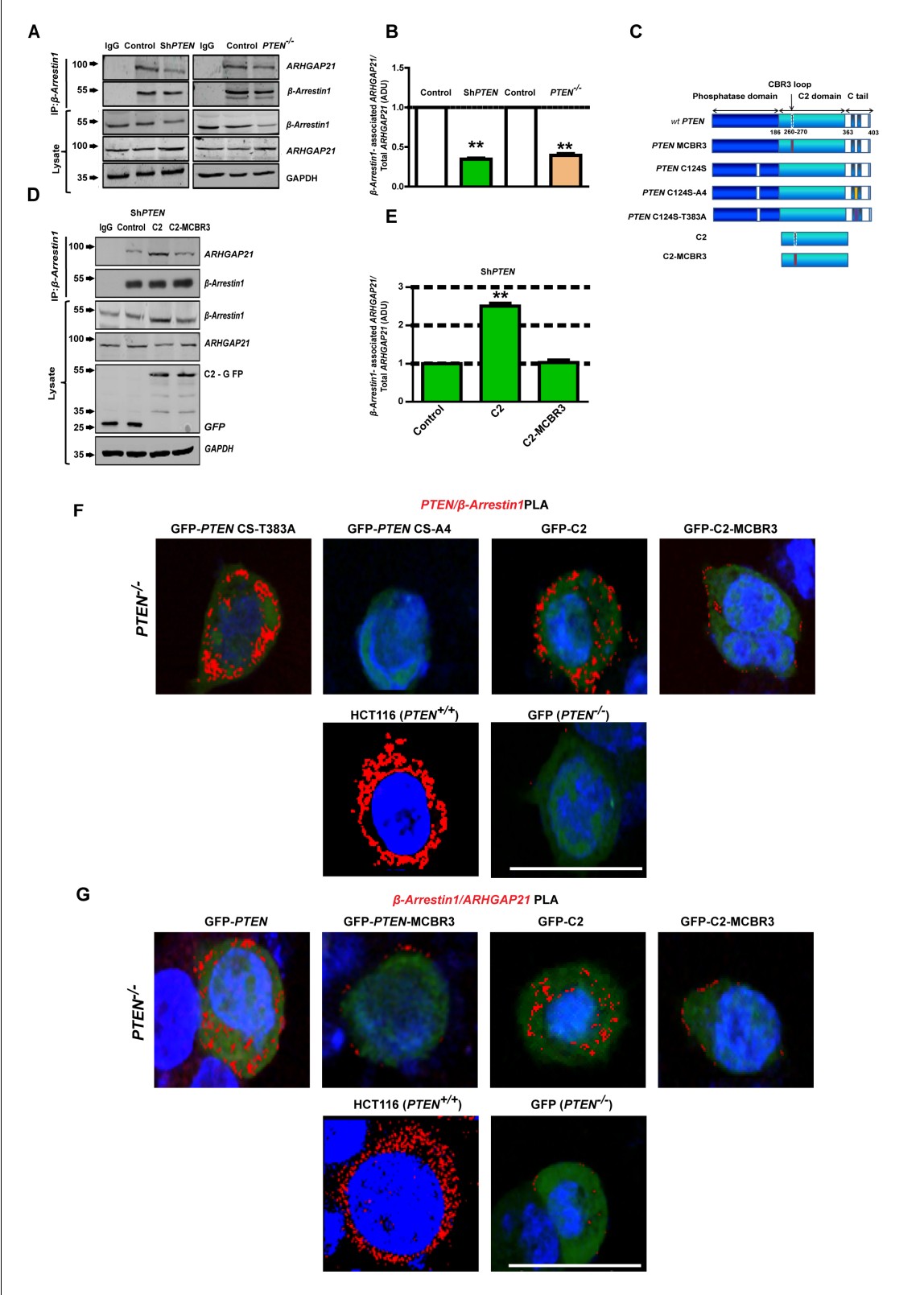

**Figure 3.** PTEN C2 enhances β-Arrestin1-ARHGAP21 binding. (**A**) β-Arrestin1-ARHGAP21 CoIPs in PTEN-expressing and -deficient cells. β-Arrestin1-associated ARHGAP21 shown in top panel against a constant β-Arrestin1 bait signal (second panel). IgG-negative controls. Total β-Arrestin1 and ARHGAP21 in lysates and *GAPDH* loading controls shown in lower three panels. (**B**) Summary β-Arrestin1-associated ARHGAP21 in PTEN-deficient (colored bars) vs PTEN-expressing cells (clear bars). Values normalized against total ARHGAP21 = 0.35 ± 0.01;**p<0.01 (Sh*PTEN*) and 0.39 ± 0.02;
*Figure 3 continued on next page*

*Figure 3 continued*

**p<0.01 (*PTEN* -/- cells), respectively. (**C**) Schematic of GFP-labeled *PTEN* constructs used (top to bottom - wild type (wt) *PTEN; PTEN* - MCBR3 membrane binding mutant; catalytically inactive *PTEN* C124S; *PTEN* C124S - A4 (CS-A4) and *PTEN* C124S -T383A (CS-T383A) mutants that lack or retain β-Arrestin1 binding capacity, respectively (*Lima-Fernandes et al., 2011*), C2 and the C2-MCBR3 membrane binding mutant. (**D**) β-Arrestin1-associated ARHGAP21 in Sh*PTEN* cells after transfection with GFP-labeled-EV control vs- C2 or -C2-MCBR3 (top panel). β-Arrestin1 bait signal shown in second panel. Total β-Arrestin1 and ARHGAP21 in lysates shown in third and fourth panels. Expression of GFP-labeled C2, C2-MCBR3 and EV and *GAPDH* loading controls shown in two lowest panels. (**E**) Summary fold change of β-Arrestin1-associated ARHGAP21 vs EV control; C2 = 2.51 ± 0.08;**p<0.01 or C2-MBCR3 = 1.03 ± 0.06; p=NS. β-Arrestin1-associated ARHGAP21 normalized against total ARHGAP21 in lysate. (**F**) Proximity ligation assay (PLA) of β-Arrestin1 interactions with PTEN constructs (red fluorescence) in *PTEN* -/- cells. Top row - GFP-labeled CS-T383A, CS-A4, C2, C2-MCBR3; Bottom row - positive control - HCT116 cells; negative control - *PTEN* -/- cells transfected with GFP only. (**G**) β-Arrestin1-ARHGAP21 interactions. Top row *PTEN* -/- cells transfected with GFP-labeled full-length *PTEN*, -*PTEN*-MCBR3, -C2 and -C2-MCBR3; Bottom row - positive control - HCT116 cells; negative control - *PTEN* -/- cells transfected with GFP only. Scale bars - 20 μm; Molecular weights indicated by arrows in blots.

DOI: https://doi.org/10.7554/eLife.24578.028

The following source data and figure supplements are available for figure 3:

**Source data 1.** Source data for *Figure 3B*.

DOI: https://doi.org/10.7554/eLife.24578.034

**Source data 2.** Source data for *Figure 3E*.

DOI: https://doi.org/10.7554/eLife.24578.035

**Figure supplement 1.** Top panel - Effects of GFP-labelled empty vector (EV) control, C2 and C2-MCBR3-GFP on β-Arrestin1-associated ARHGAP21 in *PTEN* -/- cells.

DOI: https://doi.org/10.7554/eLife.24578.029

**Figure supplement 2.** Fold changes of β-Arrestin1-associated ARHGAP21 shown in *Figure 3—figure supplement 1* vs EV control.

DOI: https://doi.org/10.7554/eLife.24578.030

**Figure supplement 2—source data 1.** *Figure 3—figure supplement 2* Transfection effects on Beta-Arrestin1-associated ARHGAP21

DOI: https://doi.org/10.7554/eLife.24578.036

**Figure supplement 3.** Diagram of Rluc-PTEN-YFP illustrating how conformational changes may alter BRET measurements, although the real orientations of donor and acceptor proteins are not known.

DOI: https://doi.org/10.7554/eLife.24578.031

**Figure supplement 4.** PTEN:β-Arrestin1 interaction AI after transfection of *PTEN* -/- cells by CS-T383A, CS-A4, C2 or C2-MCBR3.

DOI: https://doi.org/10.7554/eLife.24578.032

**Figure supplement 4—source data 1.** *Figure 3—figure supplement 4* PLA analysis of PTEN:Beta-Arrestin1 interactions.

DOI: https://doi.org/10.7554/eLife.24578.037

**Figure supplement 5.** β-Arrestin1-ARHGAP21 interaction AI after transfection of *PTEN* -/- cells by GFP-labelled C2, C2-MCBR3, wt *PTEN* or *PTEN*-MCBR3.

DOI: https://doi.org/10.7554/eLife.24578.033

**Figure supplement 5—source data 1.** *Figure 3—figure supplement 5* PLA assay of Beta-Arrestine1:ARHGAP21 interactions.

DOI: https://doi.org/10.7554/eLife.24578.038

consistent with differences in β-Arrestin1-binding capacity. We further investigated protein-protein interactions in vivo using sensitive proximity ligation assays [PLA] (*Söderberg et al., 2006*). We expressed *PTEN* phosphatase-dead mutants or C2 domain constructs in *PTEN* -/- cells. We found prominent PLA signals for PTEN-β-Arrestin1 interactions in *PTEN* -/- cells expressing either *PTEN* CS-T383A or the intact C2 domain. Conversely, *PTEN* -/- cells expressing *PTEN* CS-A4 or C2-MCBR3 mutants showed markedly reduced levels of these interaction signals. PTEN-β-Arrestin1 interaction PLA signals in HCT116 and GFP-only transfected *PTEN* -/- cells were used as positive and negative controls, respectively (*Figure 3F*; *Figure 3—figure supplement 4*). These findings indicate that PTEN-β-Arrestin1 interactions can occur independently of PTEN phosphatase activity. Next, we investigated effects of PTEN on β-Arrestin1-ARHGAP21 interactions. Transfection of *PTEN* -/- cells with GFP-labeled-wt *PTEN* or -C2 domain enhanced β-Arrestin1-ARHGAP21 interactions compared to cells transfected with *PTEN*-MCBR3 or C2-MCBR3. β-Arrestin1-ARHGAP21 interaction signals in HCT116 or GFP-only transfected *PTEN* -/- cells were used as positive and negative controls, respectively. (*Figure 3G*; *Figure 3—figure supplement 5*). Collectively, these data implicate the PTEN C2 domain in phosphatase-independent binding of β-Arrestin1 and in promoting β-Arrestin1-ARHGAP21 interactions.

## PTEN promotes β-Arrestin1 and ARHGAP21 membrane recruitment through its C2 domain

To explore effects of the isolated PTEN C2 domain on membrane recruitment of β-Arrestin1 or ARHGAP21, we conducted expression, fractionation and CoIP assays in *PTEN*-deficient cells. Expression of the C2 domain enhanced total β-Arrestin1 and suppressed that of ARHGAP21 in *PTEN*-deficient cell lysates while C2-MCBR3 had no significant effects (*Figure 4A–C*; *Figure 4—figure supplements 1–3*). C2 expression also enriched β-Arrestin1 and suppressed ARHGAP21 in membrane fractions of *PTEN*-deficient colorectal cell lines (*Figure 4D–F*; *Figure 4—figure supplement 4*). Membrane fraction values were normalized against total expression of each protein in lysate. Furthermore, expression of C2 but not C2-MCBR3 also increased β-Arrestin1-associated ARHGAP21 levels in *PTEN*-deficient cell membrane fractions. β-Arrestin1-associated ARHGAP21 levels were normalized against total ARHGAP21 in the membrane fraction (*Figure 4G–J*). In *PTEN* $^{-/-}$ cells, expression of *C2* and full-length *PTEN* had greater effects on β-Arrestin1-associated ARHGAP21 levels than *PTEN*-MCBR3, C2-MCBR3 or control (*Figure 4I,J*). *PTEN*-MCBR3 had small but significant effects on β-Arrestin1-associated ARHGAP21 levels in excess of control (*Figure 4I,J*). Taken together, these data indicate that PTEN enhances β-Arrestin1 membrane recruitment and β-Arrestin-ARHGAP21 interactions, predominantly through its membrane-binding C2 domain.

## PTEN controls mitotic spindle orientation and 3D morphogenesis by regulation of β-Arrestin1

To investigate PTEN coordination of morphogenic processes through its C2 domain, we conducted 3D organotypic culture studies. We found greater expression and membrane localization of β-Arrestin1 in 3D control PTEN-expressing Caco-2 cultures compared to Sh*PTEN* cultures (*Figure 5A*, *Figure 5—figure supplement 1*), in agreement with our biochemical analysis. Conversely, ARHGAP21 immunoreactivity was lower in control Caco-2 compared to Sh*PTEN* 3D cultures (*Figure 5B*; *Figure 5—figure supplement 2*). We have shown previously that the abnormal Sh*PTEN* 3D phenotype can be rescued by expression of the *PTEN* C2 domain (*Jagan et al., 2013b*). Here, we show that the GFP-tagged *PTEN* C2 domain enhances β-Arrestin1 membrane enrichment (*Figure 5C,D*), rescues mitotic spindle orientation (*Figure 5E,F*) as well as apical membrane alignment and single lumen morphology (*Figure 5E*; *Figure 5—figure supplements 3* and *4*) in 3D Sh*PTEN* cultures. These effects were not observed in Sh*PTEN* cultures expressing control GFP or C2-MCBR3-GFP (*Figure 5C–F*; *Figure 5—figure supplements 3* and *4*). To investigate any potential for *PTEN* ShRNA off-target effects, we investigated effects of full-length ShRNA-resistant *PTEN* (ShR *PTEN*) on the integrated Sh*PTEN* 3D morphology phenotype. We show that expression of ShR *PTEN* rescued defective morphogenesis of 3D Sh*PTEN* cultures (*Figure 5—figure supplements 5* and *6*). Collectively, these studies show that the membrane-bound PTEN C2 domain coordinates multicellular gland assembly by β-Arrestin1 membrane recruitment, mitotic spindle orientation, apical membrane alignment and lumen formation.

## PTEN controls 3D Caco-2 morphogenesis by noncatalytic coupling of β-Arrestin1, ARHGAP21 and Cdc42

Precise spatiotemporal coordination of Cdc42 activity is central to multicellular morphogenesis (*Meitinger et al., 2013*). To investigate PTEN non-catalytic regulation of Cdc42 via β-Arrestin1 and ARHGAP21, we conducted transfection and peptide inhibitor studies. Expression of the catalytically inactive *PTEN* CS-T383A construct that binds β-Arrestin1 but not the *PTEN CS-A4* binding-defective mutant, enhanced Cdc42-GTP levels in *PTEN* $^{-/-}$ cells (*Figure 6A,B*). While Cdc42 can be inhibited by ARHGAP21 (*Dubois and Chavrier, 2005*), competitive β-Arrestin1 binding to the GAP domain can release the active GTPase from ARHGAP21 inhibition (*Anthony et al., 2011*). To investigate the specific role of β-Arrestin1-ARHGAP21 interactions on Cdc42-dependent 3D morphogenesis, we used a cell-permeant 24-mer peptide analogue of the ARHGAP21 GAP domain that was designed to disrupt the β-Arrestin1-ARHGAP21 interaction (*Figure 6C*) (*Anthony et al., 2011*). Here, we show that treatment with this β-Arrestin1-ARHGAP21 peptide binding inhibitor (pep24) but not a scrambled control peptide attenuated the association between β-Arrestin1 and ARHGAP21, resulting in lower levels of β-Arrestin1-associated ARHGAP21 (*Figure 6D,E*; *Figure 6—figure supplements 1* and *2*). Treatment by pep24 also suppressed Cdc42 activation in Caco-2 (*Figure 6F,G*) and HCT116 cells

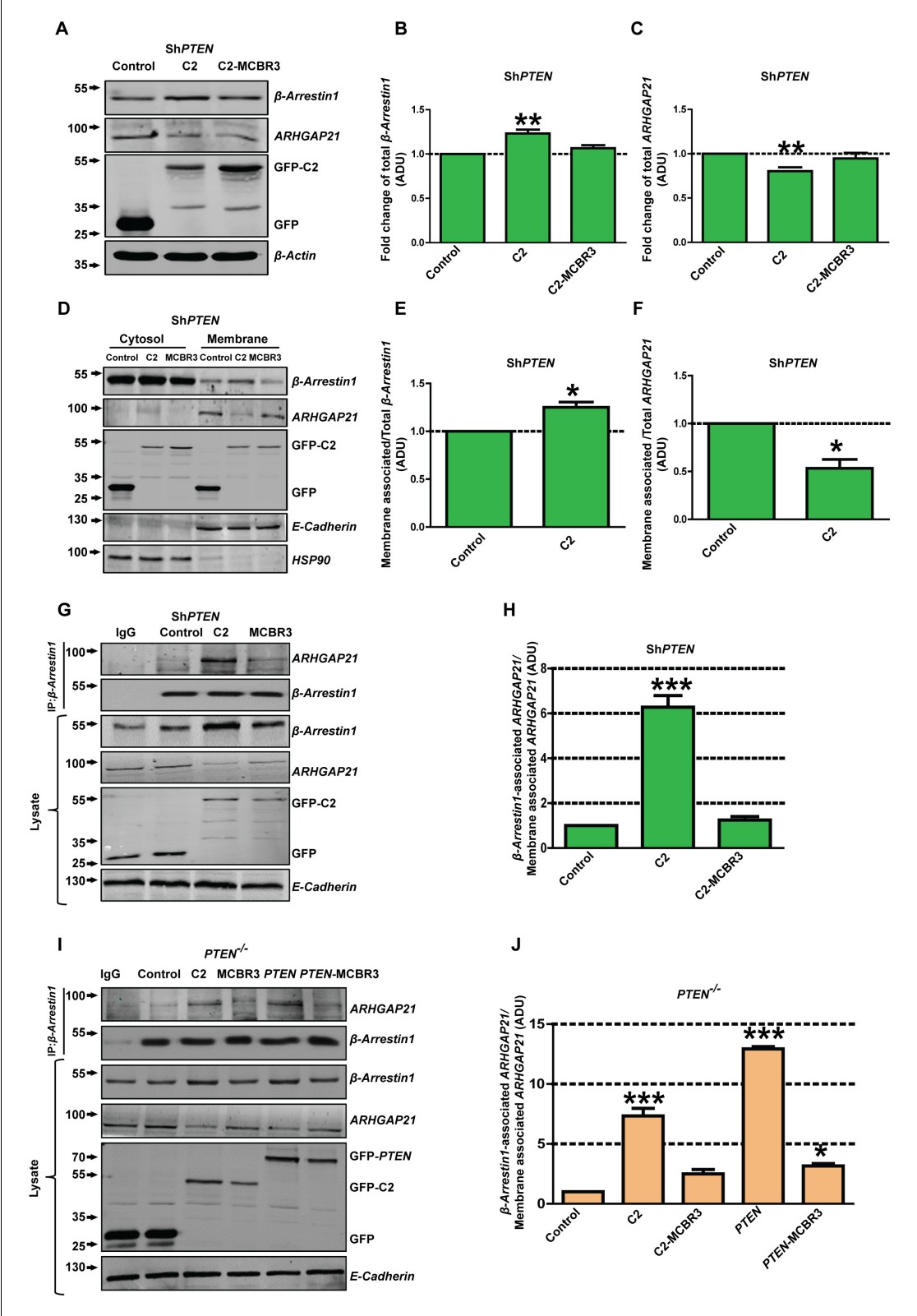

**Figure 4.** PTEN regulation of juxtamembrane β-Arrestin1 and ARHGAP21. (**A**) shows total β-Arrestin1 and ARHGAP21 expression in Caco-2 Sh*PTEN* cell lysates (top two panels) after *PTEN* C2 (**C2**), C2-MCBR3 or EV control transfections. Expression levels of transfected GFP-labelled proteins and β-Actin loading controls shown in lower two panels. (**B**) indicates summary fold changes of total β-Arrestin*1;* C2 = 1.23 ± 0.02; **p<0.01; MCBR3 = 1.07 ± 0.02; [MCBR3 vs control = NS]. (**C**) shows respective fold changes of ARHGAP21; C2 = 0.8 ± 0.02; **p<0.01; MCBR3 = 0.95 ± 0.03; [MCBR3 vs control = NS;
*Figure 4 continued on next page*

*Figure 4 continued*

all values in ADU]. (**D**) Effects of transfections on cytosol and membrane β-Arrestin1 and ARHGAP21 in Caco-2 *ShPTEN* (Sh*PTEN*) cells (top two panels). Expression levels of transfected GFP-labelled proteins and E-Cadherin and HSP90 membrane and cytosolic markers shown in lower three panels. (**E, F**) Fold change of β-Arrestin1 and ARHGAP21 expression induced in the membrane fraction. (**E**) β-Arrestin1, C2 = 1.25 ± 0.05; *p=0.04 and (**F**) ARHGAP21 = 0.53 ± 0.09; *p=0.03. β-Arrestin1 and ARHGAP21 ADU values normalized against total lysate ADU for each protein. (**G**) β-Arrestin1-associated ARHGAP21 induced in Sh*PTEN* membrane fractions by C2, C2-MCBR3 vs EV control transfections (top panel). β-Arrestin1 bait signal shown in second panel. Total β-Arrestin1 and ARHGAP21 in lysates shown in third and fourth panels. Expression of GFP-labeled C2, C2-MCBR3, EV and E-Cadherin membrane marker shown in lower three panels. (**H**) Fold changes of β-Arrestin1-associated ARHGAP21 normalized against total ARHGAP21 in the membrane fraction (ADU) - C2 = 6.27 ± 0.51; ***p<0.001; MCBR3 = 1.25 ± 0.15; [MCBR3 *vs* control = NS]. (**I**) β-Arrestin1-associated ARHGAP21 ADU after expression of C2, C2-MCBR3, wt *PTEN* or *PTEN*-MCBR3 vs EV control in *PTEN* $^{-/-}$ cell membrane fractions (top panel). β-Arrestin1 bait signal, total lysate expression of each protein, expression of GFP-labelled EV or C2 domain constructs and E-Cadherin membrane marker shown in lower five panels. (**J**) Summary fold changes of β-Arrestin1-associated ARHGAP21 normalized against total membrane ARHGAP21; C2 = 7.33 ± 0.64; C2-MCBR3 = 2.5 ± 0.35; *PTEN* = 12.93 ± 0.19; *PTEN* MCBR3 = 3.15 ± 0.21 ADU; control vs C2 or *PTEN*, ***p<0.001; control vs C2-MCBR3 (NS). Control vs *PTEN*-MCBR3 = *p<0.05. Analyses by ANOVA, Tukey post hoc or Student's paired test. Molecular weights indicated by arrows in blots.

DOI: https://doi.org/10.7554/eLife.24578.039

The following source data and figure supplements are available for figure 4:

**Source data 1.** Source data for *Figure 4B*.
DOI: https://doi.org/10.7554/eLife.24578.046
**Source data 2.** Source data for *Figure 4C*.
DOI: https://doi.org/10.7554/eLife.24578.047
**Source data 3.** Source data for *Figure 4E*.
DOI: https://doi.org/10.7554/eLife.24578.048
**Source data 4.** Source data for *Figure 4F*.
DOI: https://doi.org/10.7554/eLife.24578.049
**Source data 5.** Source data for *Figure 4H*.
DOI: https://doi.org/10.7554/eLife.24578.050
**Source data 6.** Source data for *Figure 4J*.
DOI: https://doi.org/10.7554/eLife.24578.051
**Figure supplement 1.** Total β-Arrestin1 and ARHGAP21 expression (top two panels) in.
DOI: https://doi.org/10.7554/eLife.24578.040
**Figure supplement 2.** Fold changes of β-Arrestin1 - C2 = 1.24 ± 0.03; *p<0.05; C2-MCBR3 = 1.11 ± 0.06 [MCBR3 *vs* control = NS].
DOI: https://doi.org/10.7554/eLife.24578.041
**Figure supplement 2—source data 1.** *Figure 4—figure supplement 2* Transfection effects on Beta-Arrestin1 in PTEN-/- cells.
DOI: https://doi.org/10.7554/eLife.24578.052
**Figure supplement 3.** Fold changes of ARHGAP21.
DOI: https://doi.org/10.7554/eLife.24578.042
**Figure supplement 3—source data 1.** *Figure 4—figure supplement 3* Transfection effects on ARHGAP21 in PTEN-/- HCT116 cells.
DOI: https://doi.org/10.7554/eLife.24578.053
**Figure supplement 4.** Fold changes of membrane ARHGAP21 (0.82 ± 0.03;**p<0.01).
DOI: https://doi.org/10.7554/eLife.24578.043
**Figure supplement 5.** Effects of transfection on cytosol and membrane β-Arrestin1 and ARHGAP21 in *PTEN* $^{-/-}$ cells (top two panels).
DOI: https://doi.org/10.7554/eLife.24578.044
**Figure supplement 5—source data 1.** *Figure 4—figure supplement 5* Membrane Beta-Arrestin1 in PTEN-/- cells - Source data.
DOI: https://doi.org/10.7554/eLife.24578.054
**Figure supplement 6.** Fold changes of membrane β-Arrestin1 (1.16 ± 0.03; *p=0.04).
DOI: https://doi.org/10.7554/eLife.24578.045
**Figure supplement 6—source data 1.** *Figure 4—figure supplement 6* - Transfection effects on Membrane ARHGAP21 in PTEN-/- cells.
DOI: https://doi.org/10.7554/eLife.24578.055

(*Figure 6—figure supplements 3* and *4*). Furthermore, pep24 treatment induced dysmorphogenesis of 3D Caco-2 cultures characterized by mitotic spindle misorientation (*Figure 6H*, *Figure 6—figure supplement 5*), apical membrane misalignment, aberrant epithelial configuration and loss of single central lumen formation (*Figure 6I*, *Figure 6—figure supplement 6*). Taken together, these data show that PTEN controls 3D morphogenesis by non-catalytic C2 domain scaffolding of β-Arrestin1-ARHGAP21 interactions and release of Cdc42 from ARHGAP21 inhibition.

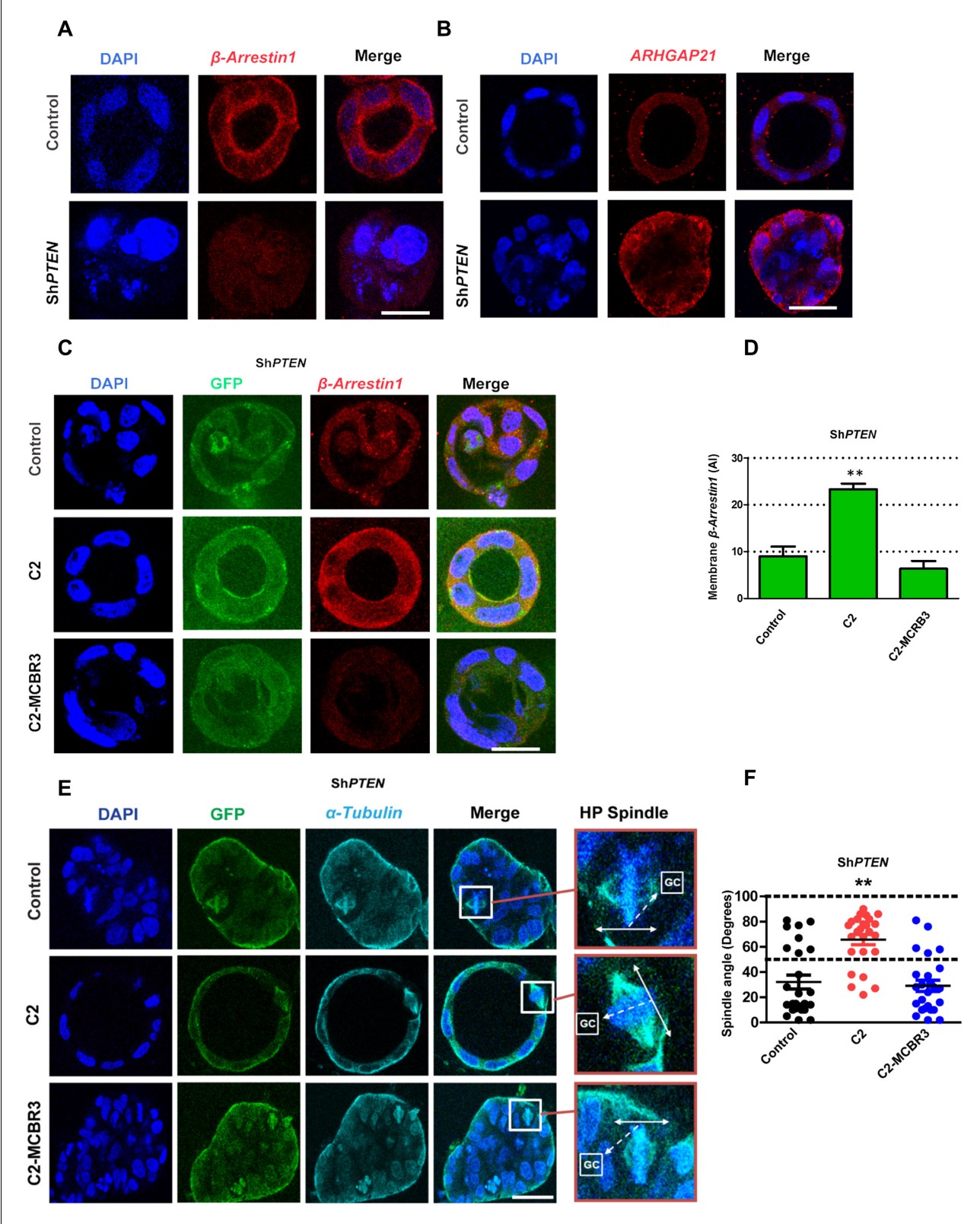

**Figure 5.** PTEN controls mitotic spindle orientation and 3D morphogenesis by noncatalytic regulation of *β-Arrestin1*. (**A**) β-Arrestin1 and (**B**) ARHGAP21 immunofluorescence intensity in 3D control Caco-2 and Sh*PTEN* cultures. (**C**) Effects of C2 vs C2-MCBR3 expression on membrane β-Arrestin1 immunoreactivity in Sh*PTEN* cultures. (**D**) Summary cell membrane β-Arrestin1 immunoreactivity AI; EV control vs C2 vs C2-MCBR3 = 9.0 ± 2.08 vs 23.3 ± 1.20 vs 6.33 ± 1.67 AI;**p<0.01, control vs C2-MCBR3 = NS. (**E**) Effects of C2 vs C2-MCBR3 expression on spindle orientation. High-power (HP)

*Figure 5 continued on next page*

*Figure 5 continued*

spindle views (orange border) enlarge areas within white rectangles and show orientation angles (interrupted white arrows) of spindle planes (double-headed solid white arrows) toward gland centres (GCs). (F) Summary spindle angles relative to GCs in 3D Sh*PTEN* cultures after transfection (• – Control = 32.08 ± 5.5⁰ vs • - C2 = 65.72 ± 4.1⁰; **p<0.01 *vs* • - MCBR3 = 29.1 ± 4.4⁰; [MCBR3 *vs* control = NS], One-way ANOVA; **p<0.01, Tukey post hoc test. Imaging - DAPI [blue], β-Arrestin1 (**A**,**C**) red, ARHGAP21, (**B**) [red], GFP [green] and α-*Tubulin* (**E**) [cyan]. Scale bars = 20 μm.
DOI: https://doi.org/10.7554/eLife.24578.056

The following source data and figure supplements are available for figure 5:

**Source data 1.** Source data for *Figure 5D*.
DOI: https://doi.org/10.7554/eLife.24578.063
**Source data 2.** Source data for *Figure 5F*.
DOI: https://doi.org/10.7554/eLife.24578.064
**Figure supplement 1.** Summary β-Arrestin1 immunoreactivity (AI) in control Caco-2 vs Sh*PTEN* organotypic cultures in *Figure 5*A = 24.3 ± 4.1 vs 11.00 ± 1.16;*p=0.04.
DOI: https://doi.org/10.7554/eLife.24578.057
**Figure supplement 1—source data 1.** *Figure 5—figure supplement 1* - Beta-Arrestin 1 intensity in Caco-2 and ShPTEN.
DOI: https://doi.org/10.7554/eLife.24578.065
**Figure supplement 2.** ARHGAP21 immunoreactivity (AI) in control Caco-2 vs Sh*PTEN* organotypic cultures in *Figure 5*B = 8.0 ± 0.58 vs 17.0 ± 2.64 AI; *p=0.02.
DOI: https://doi.org/10.7554/eLife.24578.058
**Figure supplement 2—source data 1.** *Figure 5—figure supplement 2* ARHGAP21 intensity in Caco-2 and ShPTEN glands.
DOI: https://doi.org/10.7554/eLife.24578.066
**Figure supplement 3.** Effects of expression C2 or C2-MCBR3 *vs* EV control on lumen formation in Sh*PTEN* cultures.
DOI: https://doi.org/10.7554/eLife.24578.059
**Figure supplement 4.** Effects of expression C2 or C2-MCBR3 vs EV control on lumen formation in Sh*PTEN* cultures.
DOI: https://doi.org/10.7554/eLife.24578.060
**Figure supplement 4—source data 1.** *Figure 5—figure supplement 4* Transfection effects on single lumen formation in ShPTEN.
DOI: https://doi.org/10.7554/eLife.24578.067
**Figure supplement 5.** Effects of ShRNA-resistant (shR) *PTEN* or EV control on lumen formation in Sh*PTEN* cultures.
DOI: https://doi.org/10.7554/eLife.24578.061
**Figure supplement 6.** Summary effects of shR *PTEN* vs EV control on single lumen formation in 3D Sh*PTEN* cultures - control = 11.33 ± 2.40%; ShR *PTEN* = 30.67 ± 2.91%; **p<0.01.
DOI: https://doi.org/10.7554/eLife.24578.062
**Figure supplement 6—source data 1.** *Figure 5—figure supplement 6* Effects of shRNA resistant PTEN on single lumen formation in ShPTEN glands.
DOI: https://doi.org/10.7554/eLife.24578.068

## β-Arrestin1-ARHGAP21 interactions are essential for self-assembly of normal colorectal organoids

To avoid any compromise of experimental interpretation by intrinsic Caco-2 cancer cell mutations, we investigated our key observations from cell culture experiments in organoids formed from primary normal murine colon cells (*Clevers, 2016*). In this study, we cultured colonic crypt progenitor epithelium in Matrigel supplemented with growth factors as previously described (*Sato et al., 2011*). By these methods, we generated colorectal organoids with appropriate mitotic spindle orientation, apical membrane alignment, luminogenesis and epithelial organization, in 3D cultures (*Figure 7A–C*). To investigate the role of β-Arrestin1-ARHGAP21 interactions on colorectal homeostasis, we assessed effects of pep24 vs control peptide treatment on 3D organoid morphogenesis. Here, we show that pep24 treatment perturbed mitotic spindle orientation, disrupted 3D glandular morphology and lumen formation in normal colorectal organoids (*Figure 7A–C*). Conversely, control peptide treatment had no discernible effect on 3D glandular morphology (*Figure 7A–C*). Collectively, these findings highlight a significant role for β-Arrestin1-ARHGAP21 interaction in multicellular morphogenesis of normal colorectal epithelium.

## Discussion

Scaffolding proteins have unique properties for assembling target molecules into cooperative networks within subcellular compartments (*Rock et al., 2013*) to control diverse biological functions

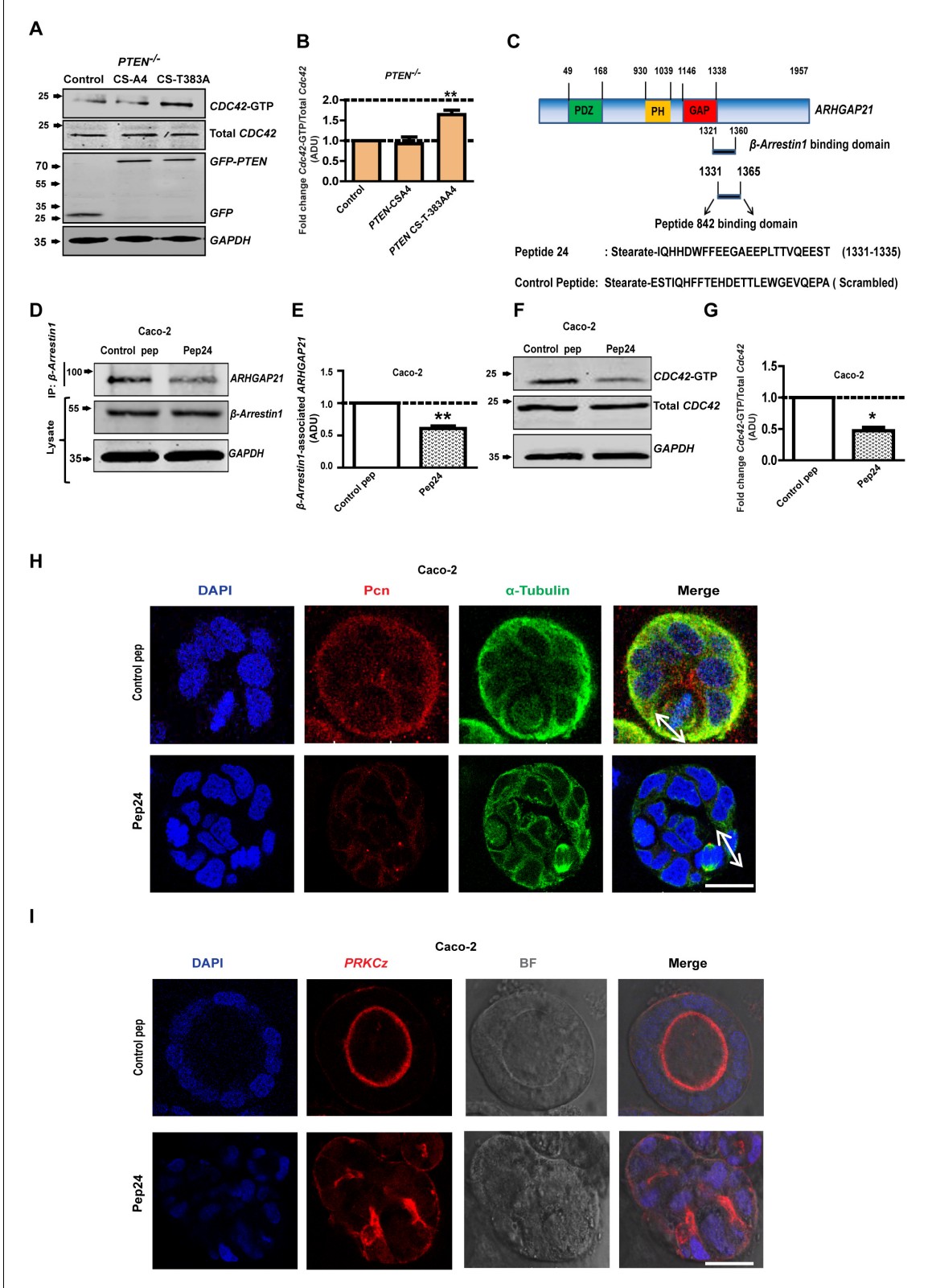

**Figure 6.** PTEN morphogenic functions mediated by β-Arrestin1, ARHGAP21 and Cdc42. (**A**) Effects of *PTEN* CS-A4 or *PTEN* CS-T383A vs EV control on *Cdc42* - GTP levels in *PTEN* $^{-/-}$ cells. (**B**) indicates fold change of Cdc42-GTP vs EV control; CS-A4 = 0.92 ± 0.16;p=NS; CS-T383A = 1.64 ± 0.11; **p=0.008. (**C**) Schematic of ARHGAP21 showing the β-Arrestin1-binding domain, the pep24 peptide-binding inhibitor and control peptide sequences (*Anthony et al., 2011*). Effects of pep24 vs control peptide (pep) on β-Arrestin1-ARHGAP21 binding (**D,E**) and Cdc42-GTP (**F,G**) in Caco-2 cells. (**E**)

*Figure 6 continued on next page*

*Figure 6 continued*

indicates fold change of β-Arrestin1-associated ARHGAP21 ADU in Caco-2 cells after pep24 vs control peptide treatment = 0.63 ± 0.02; **p=0.003. (**G**) indicates fold change of Cdc42-GTP in Caco-2 cells after pep24 vs control peptide treatment = 0.48 ± 0.05; **p=0.01. Treatment effects on spindle orientation (**H**) and lumenogenesis (**I**) in 3D Caco-2 cultures. Imaging for pericentrin (*PCN*) [red]; PRKCZ [red]; α-Tubulin [green], DAPI for nuclear DNA [blue] and bright-field (BF) imaging of lumen outlines. Spindle planes indicated by double-headed white arrows in Merge (**H**). Analyses by ANOVA, Tukey's post hoc test. Scale bar - 20 μm. Molecular weights indicated by arrows in blots.

DOI: https://doi.org/10.7554/eLife.24578.069

The following source data and figure supplements are available for figure 6:

**Source data 1.** . Figure 6B -Cdc42-GTP after transfection - source data
DOI: https://doi.org/10.7554/eLife.24578.076
**Source data 2.** *Figure 6E* - Peptide inhibitor treatment effects on Beta Arrestin1:ARHGAP21 binding in Caco-2
DOI: https://doi.org/10.7554/eLife.24578.077
**Source data 3.** *Figure 6G* Peptide inhibitor treatment effects on Cdc42-GTP in Caco-2
DOI: https://doi.org/10.7554/eLife.24578.078
**Figure supplement 1.** pep24 vs control peptide treatment effects on β-Arrestin1-associated ARHGAP21 ADU = 0.39 ± 0.09;*p=0.02.
DOI: https://doi.org/10.7554/eLife.24578.070
**Figure supplement 2.** pep24 vs control peptide treatment effects on β-Arrestin1-associated ARHGAP21 ADU = 0.39 ± 0.09;*p=0.02.
DOI: https://doi.org/10.7554/eLife.24578.071
**Figure supplement 2—source data 1.** *Figure 6—figure supplement 2* Effects of peptide binding inhibitor on Beta-arrestin1:ARHGAP21 interactions in HCT116 cells
DOI: https://doi.org/10.7554/eLife.24578.079
**Figure supplement 3.** pep24 vs control peptide treatment effects on Cdc42-GTP ADU = 0.44 ± 0.02; **p<0.01 in HCT116 cells.
DOI: https://doi.org/10.7554/eLife.24578.072
**Figure supplement 4.** pep24 vs control peptide treatment effects on Cdc42-GTP ADU = 0.44 ± 0.02; **p<0.01 in HCT116 cells.
DOI: https://doi.org/10.7554/eLife.24578.073
**Figure supplement 4—source data 1.** *Figure 6—figure supplement 4* - Peptide inhibitor treatment effects in Cdc42-GTP in HCT116 cells - source data
DOI: https://doi.org/10.7554/eLife.24578.080
**Figure supplement 5.** Summary treatment effects on mitotic spindle angles [•Control peptide = 67.4 ± 5.5⁰ *vs* • pep24 = 28.1 ± 5.1⁰; ***p<0.001] and % glands with single central lumens in 3D Caco-2 cultures (Control peptide = 35.33 ± 1.76% *vs* pep24 = 20.67 ± 1.76%;**p<0.01).
DOI: https://doi.org/10.7554/eLife.24578.074
**Figure supplement 5—source data 1.** *Figure 6—figure supplement 5* - Peptide inhibitor treatment effects on spindle angles in Caco-2 cultures
DOI: https://doi.org/10.7554/eLife.24578.081
**Figure supplement 6.** Summary treatment effects on mitotic spindle angles [•Control peptide = 67.4 ± 5.5⁰ *vs* • pep24 = 28.1 ± 5.1⁰; ***p<0.001] and % glands with single central lumens in 3D Caco-2 cultures (Control peptide = 35.33 ± 1.76% *vs* pep24 = 20.67 ± 1.76%;**p<0.01).
DOI: https://doi.org/10.7554/eLife.24578.075
**Figure supplement 6—source data 1.** *Figure 6—figure supplement 6* Peptide inhibitor treatment effects on spindle angles in Caco-2 cultures
DOI: https://doi.org/10.7554/eLife.24578.082

(*Oh and Schnitzer, 2001*; *Eroglu et al., 2003*; *Irazoqui et al., 2003*; *Smith and Scott, 2013*). β-Arrestin1 acts as a molecular scaffold for G-protein-coupled receptors [GPCRs] (*Luttrell et al., 1999*), the largest family of signaling receptors. Key GPCRs enhance β-Arrestin1 recruitment to the plasma membrane (*Urs et al., 2005*; *Li et al., 2009*; *Décaillot et al., 2011*), activate PTEN (*Song et al., 2009*; *Sanchez et al., 2005*) and promote PTEN-β-Arrestin1 interactions (*Lima-Fernandes et al., 2011*). β-Arrestin1 also suppresses ARHGAP21 (*Anthony et al., 2011*) that is independently recruited to the plasma membrane by ADP-ribosylation factor 1 [ARF-1] (*Kumari and Mayor, 2008*). In this study, we investigated PTEN coregulation of β-Arrestin1 and ARHGAP21. We found greater β-Arrestin1 levels in lysates of PTEN-expressing HCT116 cells than in the isogenic *PTEN*-null (*PTEN* ⁻/⁻) subclone and near-significant differences in corresponding PTEN-expressing and -deficient Caco-2 cells. While the precise mechanisms of this effect remain unclear, *PTEN* mutation or deficiency and changes in β-Arrestin1 expression levels characterize various human cancers (*Cantley and Neel, 1999*; *Enslen et al., 2014*).

As well as protein abundance, stoichiometry and post-translational targeting machinery modulate the assembly of spatially restricted scaffolding complexes (*Boisvert et al., 2012*). LPAR is a lysophosphatidic acid (LPA) activated GPCR that couples heterotrimeric G proteins, to control membrane recruitment of β-Arrestin1 (*Urs et al., 2005*; *Li et al., 2009*), GTPase activity (*Ueda et al.,*

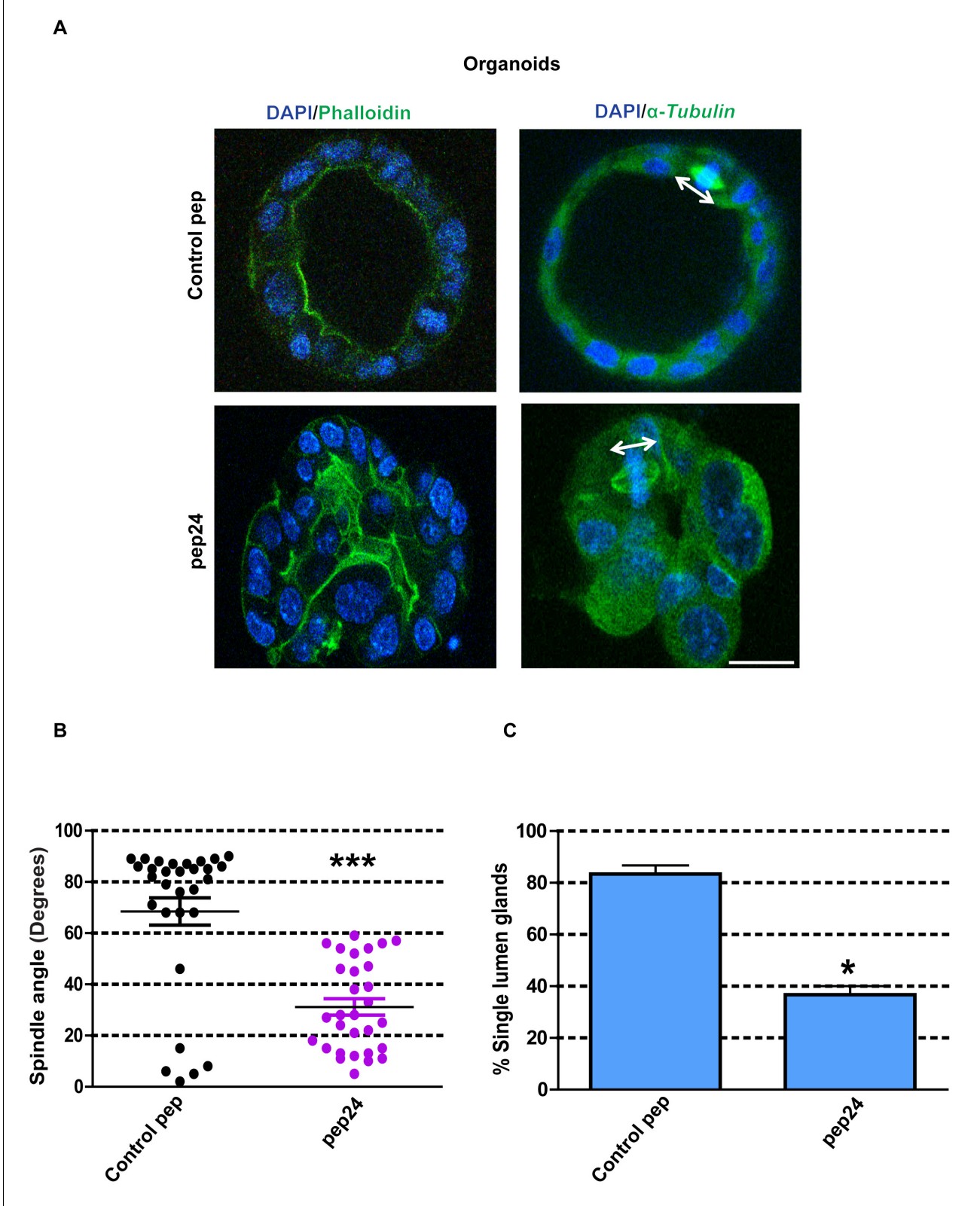

**Figure 7.** β-Arrestin1-ARHGAP21 binding is essential for morphogenesis of normal colorectal organoids. (**A**) Effects of pep24 vs control peptide (pep) on multicellular morphogenesis of normal colorectal epithelium. Imaging - apical actin marker FITC-labeled phalloidin [green], DAPI for nuclear DNA [blue] and FITC-labeled α-Tubulin [green]. Spindle orientation indicated by double-headed white arrows. (**B**) Summary spindle angles in 3D colorectal organoids after treatment (n = 30 per treatment group) •Control peptide = $68.47 \pm 5.37^0$ vs •pep24 = $31.13 \pm 3.20^0$; ***p<0.001. (**C**) Summary data

*Figure 7 continued on next page*

*Figure 7 continued*

represent percentage glandular colorectal organoids with single central lumens after treatment; control peptide = 80.00 ± 5.77% vs pep24 = 36.67 ± 3.3%;*p=0.039; (n = 10 organoids per treatment group in triplicate). Student's t test. Scale bar 20 μm.

DOI: https://doi.org/10.7554/eLife.24578.083

The following source data is available for figure 7:

**Source data 1.** Figure 7B Spindle angles in organoids - souce data.
DOI: https://doi.org/10.7554/eLife.24578.084
**Source data 2.** Figure 7C Single central lumen formation in organoids - source data.
DOI: https://doi.org/10.7554/eLife.24578.085

*2001*) and cell polarization processes (*Nagasaki and Gundersen, 1996*). To investigate these phenomena, we assessed spontaneous and LPA-mediated cell membrane localization of β-Arrestin1 and ARHGAP21 in subcellular fractions of PTEN-expressing and -deficient cells. We found proportionately greater differences of constitutive and LPA-mediated β-Arrestin1 membrane localization in PTEN-expressing Caco-2 vs PTEN-deficient Sh*PTEN* cells, indicative of PTEN involvement in β-Arrestin1 membrane recruitment. Similarly, LPA promoted greater β-Arrestin1 membrane localization in PTEN-expressing HCT116 cells than in $PTEN^{-/-}$ cells. In both cell types, we found reciprocal differences of ARHGAP21 membrane localization, consistent with β-Arrestin1-mediated suppression (*Anthony et al., 2011*). To investigate PTEN effects on β-Arrestin1 plasma membrane recruitment in whole cells, we used confocal microscopy to track the spatial distribution of transfected β-Arrestin1-mCherry against plasma membrane localization of Alexa 488-labeled WGA. Line scans of fluorescence intensities across image focal planes and high-resolution confocal z-stack reconstructions (*Furia et al., 2014*) revealed that PTEN enhanced LPA-mediated β-Arrestin1-mCherry colocalization with Alexa 488 -labelled WGA at the plasma membrane. LPA treatment had limited effects on *β-Arrestin1*-mCherry plasma membrane recruitment in Sh*PTEN* cells that have residual low level PTEN but was ineffective in $PTEN^{-/-}$ cells. We cannot attribute these findings to nonspecific fluorophore diffusion, since mCherry distribution was unaffected by PTEN status or LPA treatment. Furthermore, we can exclude PTEN nonspecific effects on cell membrane protein localization because $1,25(OH)_2D_3$ induced plasma membrane recruitment of E-Cadherin (*Pálmer et al., 2001*) was equivalent in PTEN-expressing and -deficient cells. Collectively, these findings show that PTEN is an essential coregulator of plasma membrane recruitment of β-Arrestin1 and of β-Arrestin1:ARHGAP21 functional interactions.

β-Arrestin1 and ARHGAP21 coregulate GTPases (*Anthony et al., 2011*) that function as a signaling hub for diverse cytokinetic processes (*Glover et al., 2008*; *Jaffe et al., 2008*). We investigated PTEN regulation of GTPase activity through β-Arrestin1-ARHGAP21 scaffolding and conducted perturbation experiments in organotypic 3D culture models that are ideal for precise, image-based assays of multiscale epithelial homeostasis. We found that siRNA knockdown (KD) of β-Arrestin1 suppressed Cdc42 activation in PTEN-expressing cells and 3D cultures while siRNA *ARHGAP21* KD had reciprocal effects by increasing Cdc42 activity in PTEN-deficient cultures. Furthermore, these perturbation experiments affected sequential layers of homeostatic controls. Suppression of β-Arrestin1 induced mitotic spindle misorientation, abnormal epithelial configuration, defective apical membrane positioning and formation of multiple lumens during assembly of PTEN-expressing 3D Caco-2 glandular structures. Conversely, *ARHGAP21* KD increased Cdc42 activation, restored spindle orientation and rescued aberrant morphogenesis of isogenic PTEN-deficient 3D Sh*PTEN* cultures. Because of the previously reported relationship between ARHGAP21 and RhoA (*Lima-Fernandes et al., 2011*), we assessed relationships between PTEN, β-Arrestin1, ARHGAP21 and RhoA. PTEN KD suppressed β-Arrestin1, enhanced ARHGAP21 and suppressed RhoA activation. Collectively, these findings indicate that PTEN orchestrates the cell division plane and apical membrane dynamics during multicellular morphogenesis by coordination of β-Arrestin1-ARHGAP21 functional interactions and GTPase activity.

In accord with the above findings, we found that PTEN enhanced β-Arrestin1-ARHGAP21 interactions. PTEN interacts with β-Arrestin1 (*Lima-Fernandes et al., 2011*), localizes to the plasma membrane (*Lee et al., 1999*) and modulates multicellular morphogenesis (*Jagan et al., 2013b*) via its C2 domain. Conversely, *PTEN* C2 domain mutations perturb multicellular patterning during

development and neoplastic progression (*Caserta et al., 2015*) by unclear mechanisms. C2 domain molecular interactions are masked by a closed PTEN intramolecular conformation (*Rahdar et al., 2009*). However, studies of the isolated C2 domain or an unmasked C2 mediated by alanine substitution at T383 [T383A] (*Raftopoulou et al., 2004*) within a *PTEN* C124S mutant construct, enables study of PTEN-phosphatase independent C2 domain molecular interactions. A C124S A4 mutant containing alanine substitutions at Ser$^{380}$, Thr$^{382}$, Thr$^{383}$ and Ser$^{385}$ suppresses β-Arrestin1 binding (*Lima-Fernandes et al., 2011*). We used these constructs together with appropriate tools for detection of protein binding or conformational change to investigate scaffolding functions. We found in CoIP studies that the isolated PTEN C2 domain enhanced β-Arrestin1-associated ARHGAP21 expression in excess of that induced by the C2-MCBR3 membrane-binding mutant. Bioluminescence energy transfer (BRET) analysis of Rluc-PTEN-YFP biosensor constructs containing *PTEN* CS-T383A or *PTEN*-CS-A4 (35) revealed different conformational dynamics consistent with differential protein binding. Proximity ligation assay (PLA) is a sensitive method for visualization of signals generated by protein-protein interactions (*Söderberg et al., 2006*). We found strong interactions between unmasked or isolated PTEN C2 domains and β-Arrestin1, in excess of that observed with the C2-MCBR3 or C124S A4 mutants. Furthermore, expression of wt PTEN or the C2 domain in *PTEN* $^{-/-}$ HCT116 cells enhanced β-Arrestin1-ARHGAP21 interaction signals, in excess of *PTEN*-MCBR3 (full-length *PTEN* mutated at the CBR3-membrane-binding loop) or C2-MCBR3. Interestingly, the *PTEN*-MCBR3 mutant had some limited scaffolding function, indicated by increased β-Arrestin1-ARHGAP21 binding in excess of control or C2-MCBR3 in PLA assays. Notwithstanding this finding, our data indicate that PTEN binds β-Arrestin1 and promotes β-Arrestin1-ARHGAP21 interactions predominantly through its intact C2 domain.

C2 domains are phospholipid and protein binding modules involved in membrane recruitment and localization of signaling molecules (*Corbalán-García and Gómez-Fernández, 2010*). Here, we show that PTEN C2 enhances β-Arrestin1 expression. While precise mechanisms remain unclear, PTEN C2 binds thioredoxin-1 (*Meuillet et al., 2004*) that regulates β-Arrestin1 in a context-dependent manner (*Jia et al., 2014*). In this study, PTEN C2 also promoted membrane enrichment of β-Arrestin1 in excess of total lysate concentrations and enhanced β-Arrestin1-ARHGAP21 interactions. Interestingly, wt PTEN promoted greater β-Arrestin1-associated ARHGAP21 expression in membrane fractions than the isolated C2 domain. Within its N-terminal domain, full-length PTEN contains a conserved polybasic phosphatidylinositol[$_{4,5}$] biphosphate (PtdIns [$_{4,5}$P$_2$]) [PIP$_2$]-binding site that participates in membrane-targeting (*Walker et al., 2004*) and could complement PTEN C2 domain-mediated interactions between scaffold complexes and membrane lipids. Similarly, in CoIP assays *PTEN*-MCBR3 promoted a small but significant increase of β-Arrestin1-associated ARHGAP21 expression in excess of C2-MCBR3. Taken together, our findings indicate that PTEN promotes β-Arrestin1-ARHGAP21 interactions predominantly through its C2 domain, although the PTEN N-terminal domain has a weak additional effect.

PTEN, β-Arrestins and GTPase-activating proteins modulate the activity of Rho GTPases CdcC42 and RhoA (*Martin-Belmonte et al., 2007*; *Anthony et al., 2011*; *Bos et al., 2007*; *Anderson et al., 2008*). Furthermore, the PTEN C2 domain has morphogenic properties (*Leslie et al., 2007*; *Jagan et al., 2013b*; *Caserta et al., 2015*). We investigated PTEN orchestration of multicellular gland assembly through its C2 domain, in organotypic culture studies. We found greater β-Arrestin1 and lower ARHGAP21 immunoreactivity in control Caco-2 vs Sh*PTEN* 3D cultures in accord with our findings in cell monolayers. Transfection of Sh*PTEN* cells with a C2 domain construct promoted β-Arrestin1 membrane localization, rescued mitotic spindle orientation, single central lumen formation and 3D multicellular morphogenesis. Conversely, expression of the membrane-binding mutant C2-MCBR3 domain did not rescue the morphology phenotype. Sh*PTEN* cells stably express *PTEN* ShRNA that targets the phosphatase domain (*Jagan et al., 2013a*). To assess *PTEN* ShRNA specificity and test for potential off-target effects, we investigated effects of full-length ShRNA-resistant *PTEN* (ShR *PTEN*) on the integrated Sh*PTEN* 3D morphology phenotype. Expression of ShR *PTEN* rescued 3D Sh*PTEN* morphogenesis and thus confirmed shRNA functional specificity. Collectively, the above data show that *PTEN* has important noncatalytic morphogenic functions mediated through its C2 domain and β-Arrestin1 membrane targeting.

To investigate Cdc42 activation by the unmasked *PTEN* C2 domain, we conducted expression studies in *PTEN* $^{-/-}$ cells. Transfection of *PTEN* CS-T383A robustly enhanced Cdc42-GTP, while the β-Arrestin1-binding defective CS-A4 mutant (*Lima-Fernandes et al., 2011*) had no effect. Subsequent

to these Cdc42 activation studies, we investigated the specific role of β-Arrestin1-ARHGAP21 interactions on Cdc42-dependent multicellular morphogenesis. We used a cell-permeant peptide analogue of the β-Arrestin1 docking site within the ARHGAP21 GAP domain (pep24) to disrupt β-Arrestin-ARHGAP21 interactions (*Anthony et al., 2011*). Pep24 treatment suppressed β-Arrestin1-associated ARHGAP21 expression, inhibited Cdc42 activation, induced spindle misalignment and aberrant morphogenesis of 3D Caco-2 cultures. These morphogenic effects phenocopied those of Cdc42 knockdown (*Jagan et al., 2013a*; *Jaffe et al., 2008*). Collectively, our findings indicate that PTEN C2 coordinates β-Arrestin1-ARHGAP21 and Cdc42-dependent multicellular morphogenesis in a 3D colorectal cancer model system.

PTEN is frequently downregulated in human colorectal (*Naguib et al., 2011*) and other cancers, even in the absence of genetic loss or mutation (*Salmena et al., 2008*). Regulatory cues may have a central role in tumorigenesis (*Song et al., 2012*). Activated GPCRs modulate β-Arrestin1 conformation (*Shukla et al., 2008*), membrane recruitment (*Urs et al., 2005*; *Li et al., 2009*; *Décaillot et al., 2011*) and PTEN-β-Arrestin1 interactions (*Lima-Fernandes et al., 2011*). By these processes, GPCRs may influence β-Arrestin1-dependent GTPase activation, cytoskeletal dynamics and neoplastic multicellular patterning. Studies in cancer cell models provide useful mechanistic data (*Hanahan and Weinberg, 2011*) but intrinsic mutations may compromise physiological relevance. Studies of 3D multicellular organoids isolated from normal tissues have provided basic insights into normal tissue morphogenesis (*Clevers, 2016*). We have previously generated intestinal crypt organoids for study of multicellular assembly, patterning and lineage commitment (*Campbell et al., 1993*; *Tait et al., 1994*; *Slorach et al., 1999*). Suppression of β-Arrestin1-ARHGAP21 binding in organoid systems by pep24 treatment perturbed spindle orientation and apical membrane alignment to induce a multilumen phenotype, surrounded by disorganized epithelium. Collectively, these findings demonstrate the importance of β-Arrestin1-ARHGAP21 interactions in control of normal colorectal multicellular architecture.

Within the PTEN C2 domain, the CBR3 loop can localize cytoplasmic PTEN to early endosomes arranged along the microtubule cytoskeleton, by binding endosomal $PIP_3$ (*Naguib et al., 2015*). Restriction of PTEN to a punctate vesicular distribution along microtubules may enable dephosphorylation of $PIP_3$ signals generated by plasma membrane receptor tyrosine kinases and parcelled in endosomes (*Naguib et al., 2015*). However, it is difficult to envisage that wide endosomal distribution of scaffolding interactions along the microtubule cytoskeleton could regulate the compartmentalized focus of GTPase activity (*Pertz, 2010*) required for control of spindle dynamics and multicellular morphogenesis (*Durgan et al., 2011*). Hence, dephosphorylation of $PIP_3$ on endosomes and scaffolding of β-Arrestin1-ARHGAP21 may represent spatiotemporally distinct PTEN tumor suppressor functions.

This study shows that β-Arrestin1-ARHGAP21 interactions represent an essential component of the PTEN morphogenic pathway and sheds light on conserved developmental mechanisms. In *C. elegans*, PTEN/DAF18 conducts nutrient-sensing through its phosphatase domain (*Ogg and Ruvkun, 1998*) in a negative feedback loop with the insulin/IGF axis (*Narbonne et al., 2015*) and casein kinase II [CKII] (*Liu Tj et al., 2001*). CKII phosphorylates PTEN to induce the closed conformation (*Rahdar et al., 2009*; *Torres and Pulido, 2001*) that suppresses plasma membrane binding (*Rahdar et al., 2009*). We show that the PTEN membrane-binding C2 domain is essential for multicellular morphogenesis. Hence, our findings may provide a rationale for PTEN multifaceted control of embryonic development by nutrient-sensing (*Ogg and Ruvkun, 1998*) and regulation of morphogenic growth (*Rouault et al., 1999*) according to the available nutrient energy balance (*Hietakangas and Cohen, 2009*). Our study also has oncological relevance, since disruption of PTEN C2 domain-mediated β-Arrestin1-ARHGAP21 interactions drive evolution of morphology phenotypes in 3D cultures that are evocative of colorectal cancer (*Deevi et al., 2016*; *Jaffe et al., 2008*). Dissection of these phenomena may yield novel targets for therapy aimed at suppression of aggressive cancer morphology phenotypes that predict early metastasis.

# Materials and methods

## Reagents and antibodies

All laboratory chemicals were purchased from Sigma-Aldrich, Dorset, England unless otherwise stated. RNAiMAX and X-tremeGENE transfection reagents were purchased from Thermofisher, Dublin, Ireland and Roche, Basel, Switzerland, respectively. Antibodies used in this study were anti-β-Actin (A5316; Sigma Aldrich, Dorset, England [RRID:AB_476743]); anti-β-Arrestin1 (ab32099; Abcam, Cambridge, UK [RRID:AB_722896]); anti-ARHGAP21 (55139-1-AP; Proteintech Manchester, UK [RRID:AB_10794449]); anti-E-Cadherin (562526; BD Biosciences, Oxford, UK [RRID:AB_11153868]); anti-GAPDH (ab8245; Abcam, Cambridge, UK [RRID:AB_2107448]); anti-GFP (ab8245; Abcam, Cambridge, UK [RRID:AB_298911 ]); anti-HSP90 (sc-7947; Santa Cruz, Dallas, Texas, USA [RRID:AB_2121235]); anti-Pericentrin (PCN:ab4448; Abcam, Cambridge, UK [RRID:AB_304461]); anti-Protein Kinase C ζ [PRKCZ] (ab51157; Abcam, Cambridge, UK [RRID:AB_882057]); anti-PTEN (ab32199; Abcam, Cambridge, UK [RRID:AB_777535]); anti-α-Tubulin (Ab7291; Abcam, Cambridge, UK [RRID: AB_2241126]); anti-Cdc42 (ab41429; Abcam, Cambridge, UK [RRID_726768]) and anti-Cdc42-GTP (26905; New East Biosciences, PA, USA, [RRID:AB_1961759]). These primary antibodies were used where appropriate in conjunction with Li-Cor IRDye 680 (anti-rabbit) [RRID:AB_621841] and IRDye 800 (anti-mouse) [RRID:AB_10793856] secondary antibodies, for use with the Li-Cor Infra-Red imaging systems (Li-Cor Biosciences, Lincoln, Nebraska, USA) in Western blots or with Alexa Fluor 568 (anti-rabbit) [RRID:AB_143011] and Alexa Fluor 488 (anti-mouse) [RRID:AB_141626;Molecular probes, Invitrogen, CA, USA] and/or anti-mouse CY5 (Jackson Immunoresearch, Newmarket, Suffolk, UK[RRID:AB_[RRID:AB_2340152]) for fluorescence or confocal microscopy. We obtained Alexa 488-labeled wheat germ agglutinin (WGA) from ThermoScientific Dublin (Product No W11261). DNA was imaged with DAPI (Vector Scientific, Belfast, NI) while FITC-labeled phalloidin (p5282; Sigma-Aldrich, Dorset, England) was used to image apical actin in organoid cultures. For PLA, studies we used mouse anti-β-Arrestin1 from ThermoScientific, Paisley, UK with Duolink in situ fluorescence kits, (Sigma-Aldrich, Dorset, England) according to manufacturer's instructions. SiRNA oligonucleotides targeted against β-Arrestin1 (Qiagen Flexitube; 1027417) or ARHGAP21 (Dharmacon SmartPool; L-009382-01-0005) or nontargeting (NT) scrambled controls were purchased from Fisher Scientific, Dublin, Ireland. The cell permeant β-Arrestin1-ARHGAP21 binding disruptor peptide [pep24 - based on amino acids 1331 to 1355 within the ARHGAP21 GAP domain (*Anthony et al., 2011*)] and scrambled control peptide were purchased from *EZ* Biolabs, Carmel, IN 46032 USA. Pep24 and control peptides were prepared in dimethyl sulfoxide (DMSO), according to manufacturer's instructions. For the pep24 experiments, cells were incubated in 2D or 3D cultures as outlined below for 48 hr, then treated with either 10 µM pep24 or 10 µM control peptide. Incubations were continued for 24 hr for assays of protein binding or Western blots in cell monolayers. In 3D morphogenesis assays, test and control peptides were added to the media in the above concentrations, changed at 48 hr intervals and effects on morphogenesis assessed at 4 days of culture.

## Cell lines

Stable *PTEN*-deficient Caco-2 Sh*PTEN* (Sh*PTEN*) cells were generated by transfection of parental Caco-2 cells (obtained from the American Type Culture Collection, Manassas, VA [RRID:CVCL_0025]) with replication-defective retroviral vectors encoding *PTEN* short hairpin RNA (ShRNA), using the Phoenix retroviral expression system (Orbigen, San Diego, CA USA), as previously described (*Jagan et al., 2013a*; *Deevi et al., 2011*). *PTEN* $^{+/+}$ HCT116 [RRID:CVCL_0291] and *PTEN* $^{-/-}$ HCT116 (here known as HCT116 and *PTEN* $^{-/-}$) colorectal epithelial cells were a gift from Dr Tod Waldman, Georgetown (*Lee et al., 2004*) and were cultured in McCoys 5A media supplemented with 10% FCS (fetal calf serum), 1 mM L-glutamine and 1 mM sodium pyruvate. Caco-2 and Sh*PTEN* cells were cultured in Dulbecco's modified Eagle's medium (DMEM) supplemented 10% FCS, 1 mM non-essential Amino Acids and 1 mM L-Glutamine at 37°C in 5% CO$_2$. In 3D cultures, Caco-2, Sh*PTEN* cells and subclones transfected with SiRNAs, *PTEN* C2 domain or empty vector (EV) control constructs, were cultured embedded in a Matrigel matrix (BD Biosciences, Oxford, UK), as previously described (*Jagan et al., 2013a*; *Jagan et al., 2013b*). Caco-2 and HCT116 cells were characterized in terms of *PTEN* expression, AKT signaling, GTPase activation (*Jagan et al., 2013a*) and Caco-2 morphogenic growth (*Jagan et al., 2013a*). Furthermore, short tandem repeat (STR) profiling

(*Capes-Davis et al., 2013*) conducted by LGC Standards, Middlesex, UK confirmed authenticity by 100% and 94% matches, respectively, between study parental Caco-2 and HCT116 cells and original American Type Culture Collection (ATCC) derivatives.

## Cell transfection

We carried out mammalian SiRNA and DNA transfections using RNAiMAX and X-tremeGENE transfection reagents respectively, according to manufacturer's protocols. Cells were plated at $2 \times 10^5$ cells/35 mm dish for 24 hr, then transfected with 10 µM siRNA or 500 ng DNA/$2 \times 10^5$ cells for all respective siRNA oligonucleotides or DNA constructs. Cells were incubated with RNA/RNAiMAX or DNA/X-tremeGENE lipofectamine complexes for 48 hr, before lysis and probing. In 2 Sh*PTEN* cells, the stably expressed *PTEN* ShRNA targets a 58 base pair region within the *PTEN* phosphatase coding region and C2 domain constructs are unaffected (*Boehm et al., 2005*). In membrane localization studies, PTEN-expressing and -deficient Caco-2 and HCT116 clones were transfected with β-Arrestin1-mCherry against mCherry only controls. In expression, co-immunoprecipitation and morphogenesis studies, Sh*PTEN* cells were transiently transfected with empty vector (EV) only, the isolated *PTEN* C2 domain (C2) or a C2 domain construct mutated at the CBR3 membrane-binding loop [C2-MCBR3] (*Lee et al., 1999*) in pEGFP expression vectors.

## *PTEN* mutants

*PTEN*-C124S-A4 and *PTEN*-C124S-T383A were generated by introduction of four alanine substitutions at Ser380, Thr382, Thr383 and Ser385 and by alanine substitution at Thr383 only, respectively (*Raftopoulou et al., 2004*) into lipid and protein phosphatase dead *PTEN* C124S (*Maier et al., 1999*). These mutants suppress or enhance β-Arrestin1 binding, respectively (*Lima-Fernandes et al., 2011*). *PTEN*-MCBR3, the isolated C2 domain and C2-MCBR3 were gifts from Dr N Leslie, Dundee and were generated by replacement of 263-K-M-L-K-K-D-K-269 in the C2 domain CBR3 membrane targeting loop with the 263-A-A-G-A-A-D-A-269 sequence (*Lee et al., 1999*), as previously described (*Jagan et al., 2013b*). Sequence specificities of C2-MCBR3 and *PTEN*-MCBR3 mutants were confirmed by sequencing studies.

## Cell fractionation

We conducted these experiments using a subcellular fractionation kit (Thermo Fisher Scientific, Dublin, Ireland) according to manufacturer's protocol. Briefly, cells were trypsinized and lysed in cytoplasmic extraction buffer for 10 min at 4°C, then centrifuged at 500 g for 5 min. The supernatant was collected as the cytoplasmic fraction while the pellet was resuspended in membrane extraction buffer, vortexed for 5 s and mixed gently for 10 min at 4°C. The mix was centrifuged at 3000 g for 5 min and the supernatant was collected as the membrane fraction. In separate experiments, we conducted protein extraction, Western blotting and co-immunoprecipitation (Co-IP) assays in isolated cell membrane and cytosolic fractions. Equivalent amounts of membrane fraction and cytosol were loaded in immunoblots and Co-Ips.

## Protein extraction and western blotting

As previously described (*Jagan et al., 2013a*; *Jagan et al., 2013b*), proteins were resolved using gel electrophoresis, followed by blotting onto nitrocellulose membranes. Membranes were probed using antibodies as indicated in the text. Experiments were repeated in triplicate.

## Co-immunoprecipitation (Co-IP)

Cells were lysed on ice in buffer containing 100 mM Tris-HCl, pH 7.5, 1% Triton X-100, 5 mM EDTA, 5 mM EGTA, 50 mM NaCl, 5 mM NaF, 1 mM $Na_3VO_4$ and protease inhibitor. Cell lysates were centrifuged (for 10 mins at 15,000 g) and protein concentrations were measured by the BCA method. 1000 µg of protein was precleared overnight with control IgG and 15 µl of Protein A/G Sepharose beads (Santa Cruz, Dallas, Texas, USA). The protein was then immunoprecipitated with the appropriate antibody-beads conjugate and incubated on a rotating wheel for 2 hr. The beads were collected by centrifugation and washed five times in wash buffer (50 mM Hepes, pH 7.4, 1% Triton X-100, 0.1%, SDS, 150 mM NaCl, 1 mM $Na_3VO_4$). The beads were subsequently resuspended in 40 µl Laemmli sample buffer and processed for gel electrophoresis.

## GST-based-GTPase pulldown assays

Experiments were conducted as previously described (*Deevi et al., 2016*; *Jagan et al., 2013b*;). Briefly, cells were grown on 90 mm dishes then lysed in buffer comprising 50 mM Tris-HCl (pH 7.5), 1% Triton X-100, 100 mM NaCl, 10 mM MgCl$_2$, 5% glycerol, 1 mM Na$_3$VO$_4$ and protease inhibitor cocktail (Roche) and centrifuged at 12,500 g for 10 min. We assayed the GTP-bound form of RhoA by adding GST-Rhotekin fusion protein coupled with gluthathione sepharose 4B beads (Sigma-Aldrich, Dorset, England) to 1 mg of cell lysate. Beads were collected after 1 hr by centrifugation, washed x3 and resuspended in Laemmli buffer with 1 mM DTT. RhoA -GTP levels were then assayed by western blotting. Experiments were repeated in triplicate.

## Proximity ligation assays

We assessed protein-protein proximities using the Duolink II red kit (Sigma-Aldrich, Dorset, UK) according to the manufacturer's instructions. Briefly, we transfected PTEN$^{-/-}$ cells with GFP tagged-EV or -PTEN constructs and cultured the cells in Millipore eight well chambers. After 24 hr, we fixed the cells with 4% paraformaldehyde (PFA) at room temperature for 20 min. We then permeabilized the cells with 0.05% TritonX100 in PBS for 10 min. Cells were blocked with immunofluorescence (IF) buffer (Duolink, Sigma-Aldrich, Dorset), England for 2 hr according to manufacturer's instructions and incubated with primary antibody overnight at 4°C. Cells were washed twice with buffer A, and incubated with PLA probe, ligase and polymerase according to the manufacturer's protocol. Finally, cells were washed with buffer B and slides were mounted with a cover slip using Duolink in situ mounting medium with DAPI.

## BRET assays

BRET investigations were performed as described previously (*Lima-Fernandes et al., 2014*). Briefly, HEK cells were transfected with the indicated plasmids 24 hr after seeding. At 24 hr post transfection, cells were detached, resuspended in full media, and distributed into poly-l-orthinine coated white 96-well optiplates (Perkin Elmer). The following day, cells were washed with PBS and then overlayed with HBSS. Coelenterazine h was added to a final concentration of 5 mM and incubated for 3 min at 25°C. BRET readings were collected using a Multimode Reader Mithras$^2$ LB 943 (Berthold Technologies). Substrate and light emissions were detected at 480 nm (Rluc) and 540 nm (YFP) for 1 s. The BRET signal was calculated by ratio of the light emitted by YFP and the light emitted by Rluc (YFP/Rluc). The ratio values were corrected by substracting background BRET signals detected when Rluc-PTEN was expressed. mBRET values were calculated by multiplying these ratios by 1000. ΔmBRET values are shown to demonstrate the shift in BRET signal compared to wt signal, which is set to zero, or between the two mutants (C124S-T383A and C124S-A4) that were tested.

## Three-dimensional (3D) cultures and morphogenesis assays

Caco-2 and Caco-2 ShPTEN cells were cultured and embedded in Matrigel matrix (BD Biosciences, Oxford, UK), then imaged by confocal microscopy as we previously described (*Jagan et al., 2013a*; *Jagan et al., 2013b*). Briefly, 6 × 10$^4$ trypsinized cells were mixed with Hepes buffer (20 mM, pH 7.4) and Matrigel (40%) in a final volume of 100 μl, placed in each well of eight-well multichambers (BD Falcon, Fisher Scientific, Dublin, Ireland), allowed to solidify for 30 min at 37°C and subsequently overlayed with 400 μl of media/well. We imaged the 3D cultures at progressive stages of morphogenesis as previously described (*Jagan et al., 2013a*; *Jagan et al., 2013b*).

## Colorectal organoid cultures

We used C57B/6 wild-type mice (1–6 weeks old) for experiments and conducted all animal procedures in accordance with local and national regulations. We isolated organoids as previously described (*Tait et al., 1994*; *Slorach et al., 1999*). Briefly, murine colons were opened longitudinally, cut into 0.5 cm fragments, washed 7–10 times in 1x HBSS (low calcium, low magnesium (Gibco-BRL), 2% D-glucose, 0.035% NaHCO$_3$) to remove all luminal contents. The fragments were then finely chopped with a scalpel and digested in HBSS solution containing collagenase and dispase I neutral proteases (Sigma-Aldrich, Dorset, UK) at 1 mg/ml for 20 min at room temperature on a shaking platform. Digestion was stopped by the addition of 30 ml DMEM/F12 culture medium (Life Technologies, Renfrew UK) supplemented with 5% FCS containing penicillin and streptomycin. Large

fragments and muscle sheets were allowed to settle to the bottom of the flask. We removed the supernatant containing the organoids and centrifuged it for 3 min at 250 rpm, to pellet the organoids. We removed the supernatant and gently resuspended the organoid pellet in 20 ml of the DMEM/F12 solution. We repeated the centrifugation step 5–6 times until the pellet contained a homogeneously sized organoid preparation. Organoids thus prepared were resuspended in a 2x volume of Matrigel (growth factor reduced, phenol red free; BD Biosciences, Oxford UK) supplemented with 50 ng/ml murine EGF, murine Noggin 100 ng/ml (PeproTech, NJ, USA) and 1 µg/ml human R-Spondin, as indicated for organoid culture (*Sato et al., 2011*). Eight well multichambers were coated with a thin layer of undiluted Matrigel and allowed to solidify. Organoid preparations in Matrigel (100 µl suspension) were placed into each well, then overlaid with 250 µL/well culture medium (Dulbecco's modified Eagle medium/F12) supplemented with penicillin/streptomycin, 10 mmol/L HEPES, Glutamax supplements 1× N2, 1 × B27 [Invitrogen], 1 mmol/L *N*-acetylcysteine [Sigma]), 50 ng/ml murine EGF, Noggin 100 ng/ml and 1 µg/ml human R-Spondin (*Sato et al., 2011*). We cultured the organoids for 4 days with peptide treatments as defined.

## Confocal immunofluorescence microscopy

Membrane and cytosolic localization of β-Arrestin1-mCherry or mCherry only were imaged against Alexa 488-labelled WGA, a widely used fluorescent marker that binds to cell membranes (*Crossman et al., 2015*) in Caco-2, Sh*PTEN*, HCT116 and *PTEN*$^{-/-}$ cells, with or without LPA treatment. We used a Leica SP8 confocal microscope and Leica LAS-X software for line scanning of fluorescent images. Caco-2 Sh*PTEN* (Sh*PTEN*) glands and organoid cultures were incubated in 4% PFA for 20 min and processed for immunofluorescence as previously described (*Jagan et al., 2013a*; *Deevi et al., 2016*; *Jagan et al., 2013b*). Briefly, 3D cultures were fixed in PFA for 20 min at room temperature and permeabilized for 10 min in 0.5% Triton X-100 in PBS. The 3D cultures were rinsed with PBS/glycine buffer for 15 min to reduce autofluorescence and blocked by incubation in IF Buffer (PBS with 0.1% bovine serum albumin, 0.2% Triton X-100, 0.05% Tween-20)+10% goat serum, for 1–1.5 hr at room temperature. Primary antibodies were diluted in blocking buffer and incubated overnight at 4°C. The 3D cultures were incubated with secondary antibodies and/or FITC-labeled phalloidin for 1 hr. DNA was stained using Vectashield mounting medium containing DAPI (Vector Scientific, Belfast, NI). Sequential scan images were taken the midsection of glands/organoids at room temperature using a Leica SP5 confocal microscope [RRID:SCR_012314] on a HCX PL APO lambda blue 63 × 1.40 oil immersion objective at 1x or 2x zoom. Images were collected and scale bars added using LAS AF confocal software (Leica) [RRID:SCR_013673]. We assessed effects of transfection or treatment on signal intensity, spindle orientation, lumen formation and/or epithelial configuration in in 3D glands or organoids at 4 days of culture. Because imaging for apical protein kinase C zeta (PRKCZ) was unsuccessful in organoids, the apical domain was imaged using FITC-labelled phalloidin as a marker of apical actin.

## Assessment of mitotic spindle orientation

In cultured cells, centrosomes (Csms) were identified using anti-pericentrin (PCN) and microtubules by anti-α-Tubulin antibodies, respectively, and we identified chromosomal DNA by DAPI staining. We defined bipolar mitotic spindle architecture by convergence of microtubules towards each of 2 spindle poles, as we previously described (*Deevi et al., 2016*). Caco-2 and Caco-2 sh*PTEN* glands were cultured in Matrigel for 4 days, fixed with 4% PFA and stained with anti α-Tubulin, PRKCZ and PCN primary antibodies. Gland midsections were imaged by confocal microscopy to identify cells containing well-formed mitotic spindles, during metaphase or anaphase. Lines connecting each spindle extremity were drawn using ImageJ and the line center was considered as the spindle midpoint. Angles between spindle planes and lines connecting spindle midpoints to gland centres were measured, as outlined previously (*Deevi et al., 2016*; *Jaffe et al., 2008*). We used a similar approach or imaging of organoid morphogenesis and identified apical domains and spindles using FITC-labeled phalloidin and anti-α-Tubulin antibodies, respectively.

## Image processing and statistical analysis

Confocal microscopy images were processed using Leica Fw4000 Imaging software and cropped using Adobe Ilustrator [RRID:SCR_014198]. Confocal images were processed, merged and mean

area quantified using LAS AF Leica imaging software, as previously described (*Jagan et al., 2013a*; *Jagan et al., 2013b*). We assessed lumen formation, spindle orientation and signal intensities using 50, 15 or 10 × 3D Caco-2 or Sh*PTEN* glandular cultures (glands) in triplicate for each experimental condition, respectively. We selected glands with mitotic figures for spindle orientation assays. Organoids were fewer in number and we assessed lumen formation and spindle orientation in 10 organoids per experimental condition, in triplicate. Multicellular structures with single central lumens were expressed as a percentage and spindle orientation angles relative to gland centres were calculated using ImageJ.

### Data analysis

Descriptive statistics were expressed as the mean ± sem. Statistical analyses were by one or two-way ANOVA with the Tukey post hoc test or Student's paired t test using Graphpad Prism software (v5; Graphpad CA 92037 USA [RRID:SCR_002798]). Scatterplots and bar charts were used for display of quantitative numerical or categorical data.

## Acknowledgements

We thank Dr Todd Waldman (Georgetown Q34 University) for supply of *PTEN* $^{+/+}$ and $^{-/-}$ HCT116 cells and the late Professor Alan Hall, Sloan Kettering, NY, USA for provision of the *PTEN* C2 domain vector and Dr Nick Leslie, Dundee for provision of C2-MCBR3 and *PTEN*-MCBR3 mutant constructs. We gratefully acknowledge Cancer Research UK (Grant Number C9136/A15342) and the Department of Education and Learning, Northern Ireland for financial support. Work in the group of MGHS is supported by Fondation ARC pour la Recherche sur le Cancer, Ligue Contre le Cancer (comité de l'Oise), LABEX, CNRS and INSERM. GSB was funded by the Medical Research Council grant MRC (MR/J007412/1).

## Additional information

### Funding

| Funder | Grant reference number | Author |
|---|---|---|
| Cancer Research UK | C9136/A15342 | Frederick C Campbell |
| Department of Education and Learning, Northern Ireland | Studentship | Arman Javadi |
| Fondation ARC pour la Recherche sur le Cancer | | Mark GH Scott |
| Medical Research Council | MRC(MR/J007412/1) | George S Baillie |

The funders had no role in study design, data collection and interpretation, or the decision to submit the work for publication.

### Author contributions

Arman Javadi, Formal analysis, Validation, Investigation, Visualization, Methodology; Ravi K Deevi, Formal analysis, Supervision, Validation, Investigation; Emma Evergren, Formal analysis, Writing—original draft; Elodie Blondel-Tepaz, Data curation, Formal analysis, Investigation; George S Baillie, Mark GH Scott, Investigation, Writing—review and editing; Frederick C Campbell, Conceptualization, Formal analysis, Supervision, Funding acquisition, Visualization, Writing—original draft, Project administration, Writing—review and editing

### Author ORCIDs

Mark GH Scott  http://orcid.org/0000-0002-1557-1856
Frederick C Campbell  http://orcid.org/0000-0002-0363-9964

### Decision letter and Author response

Decision letter https://doi.org/10.7554/eLife.24578.088

Author response https://doi.org/10.7554/eLife.24578.089

## Additional files

### Major datasets

The following dataset was generated:

| Author(s) | Year | Dataset title | Dataset URL | Database, license, and accessibility information |
|---|---|---|---|---|
| Javadi A, Deevi RK, Evergren E, Blondel-Tepaz E, Baillie GS, Scott MGH, Campbell FC | 2017 | Data from: PTEN controls glandular morphogenesis through a juxtamembrane β-Arrestin1/ARHGAP21 scaffolding complex | https://dx.doi.org/10.5061/dryad.ns5qs | Available at Dryad Digital Repository under a CC0 Public Domain Dedication |

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
