## [Decision Letter]

[Editors’ note: a previous version of this study was rejected after peer review, but the authors submitted for reconsideration. The first decision letter after peer review is shown below.]

Thank you for submitting your work entitled "PTEN controls glandular morphogenesis through a juxtamembrane β-Arrestin1/ARHGAP10 scaffolding complex" for consideration by *eLife*. Your article has been reviewed by four peer reviewers, and the evaluation has been overseen by Mohan Balasubramanian as Reviewing Editor and a Senior Editor. The following individuals involved in review of your submission have agreed to reveal their identity: Michel Bagnat (Reviewer #4).

Our decision has been reached after consultation between the reviewers. Based on these discussions and the individual reviews below, we regret to inform you that your work will not be considered further for publication in *eLife*.

The reviewers and the editors found the premise of your study, of a non-catalytic function of PTEN in morphogenesis via modulation of β-arrestin dynamics very interesting. While the reviewers were positive about your study, in discussions it emerged that all of them agreed that the biochemical experiments raised important concerns.

In light of this, we urge you and your colleagues to carefully address the concerns raised, by all four reviewers. In particular, reviewer 1 has detailed the concerns in all the 7 points this reviewer has raised. Some of these points have raised by each of the other three reviewers.

Our decision is to turn down the submission, as we do not know if your current conclusions will stand after the experiments we suggest are completed. If the conclusions still hold up, we will be happy to review a new manuscript from you, should you wish to submit to *eLife*.

Reviewer #1:

In this paper, Javadi et al. describe their investigation of the function of the tumor suppressor PTEN in morphogenesis and spindle orientation in 3D multicellular organoids. They concluded that the loss of PTEN or β arrestin causes impairment in both these functions and that these defects are restored by activation of Cdc42 through suppressing its GTPase activation protein, ARHGAP10. They also suggest that PTEN controls the interaction between β arrestin and ARHGAP10 through its C2 domain, and that disruption of this interaction leads to the inhibition of Cdc42. The data for the 3D organoid formation and the regulation for spindle orientation are very interesting and convincing. However, this reviewer has several problems with the presentation and reproducibility of the data, which makes scientific interpretation difficult. I believe the authors should meticulously review their manuscript and remedy the problems in this otherwise potentially very interesting and important work. Specifically:

1) In Figure 1, the authors show that knockout or knockdown of PTEN leads to decreased membrane-bound β arrestin levels and increased membrane-bound ARHGAP10 levels, as evidenced by a crude membrane fractionation assay. However, the amounts of membrane-bound β arrestin are inconsistent between Figure 1 (showing a 50% decrease) and Figure 1 (showing almost no decrease). To compare each group shown in Figure 1, statistical analysis was performed only by ANOVA without multiple comparisons. Furthermore, for the same experiment shown in Figure 1—figure supplement 3, Western blotting does not show a decrease in membrane-bound β arrestin levels in PTEN KO, although the quantification shows a 20% decrease at the basal level (before LPA stimulation). Again, statistical analysis is incomplete.

2) The intracellular localization of β arrestin is shown in Figure 1 and Figure 1—figure supplement 1. The images are improperly modified; the size and cropped area are not the same in each image. Also, in each case, the image quality is very low and pixelated, making it difficult to evaluate their plasma membrane localization.

3) In Figure 1, the authors suggest that in the PTEN knockdown, the β arrestin level in the membrane fraction decreases while the ARHGAP10 level increases. They also show in Figure 4 that expression of the C2 domain of PTEN restores these changes caused by PTEN KD/KO. There is no ARHGAP10 in the cytosol fraction. The authors should include essential control to show that the total amounts of β arrestin and ARHGAP10 are not affected by PTEN KD/KO. What percentage of cytosolic and membrane proteins are loaded?

4) In a related note, the loss of PTEN may simply change the expression levels of β arrestin and ARHGAP10 and not their membrane localizations. This possibility is supported by many data (intensity of β arrestin in Figure 1, Western data for total β arrestin expression in cells expressing C2 in Figure 3, and increased intensity of β arrestin and decreased intensity of ARHGAP10 in C2-expressing 3D organoids in Figure 5).

5) In Figure 3, the authors show that loss of PTEN decreases β arrestin-ARHGAP10 interaction. Ectopic expression of the C2 domain, but not the MCBR3 mutant, rescues this phenotype. The expression levels of C2 and MCBR3 should be examined to exclude the possibility that they have different expression levels and that is the reason for their distinct rescuing activities. They should also test whether C2 or MCBR3 interacts with the β arrestin-ARHGAP10 complex.

6) In Figure 4 through F, the authors show that expression of C2 leads to increases in the amounts of membrane-bound β arrestin and decreases in membrane-bound ARHGAP10 levels in PTEN KD/KO cells. How much C2 and MCBR3 are associated with the membrane fraction? Again, the authors need to show the total expression levels of these proteins.

7) The increase in β arrestin in the membrane fraction by the expression of C2 shown in Figure 4 is not observed in Figure 4 and I.

Reviewer #2:

Previous studies have shown that b-arrestin interacts with ARHGAP10 and negatively regulates its GAP activity. Here the authors show that PTEN positively regulates membrane localization of b-arrestin. Consistent with previously reported findings the authors show that the C2 domain of PTEN mediates the interaction with b-arrestin. In addition, the author investigate the biological significance of this pathway using caco2 3D culture system and investigate changes in spindle orientation during cell division and its effect of morphogenesis. This is an interesting study that connects PTEN with ARHGAP10 pathways and adds more insight into the previously known relationship between PTEN and b-arrestin. However, as outlined below, additional insights into the relationship between PTEN and b-arresting/ARHGAP10 interaction would increase the impact of the study.

Figure 1 – the authors state loss of PTEN results in "greater b-arresting but lower ARHGAP10 levels in membrane fractions" but the data presented show a lower b-arrestin and greater ARHGAP10.

Figure 1: Can the authors show a control membrane protein that can translocate to membrane upon a ligand stimulation irrespective of PTEN levels. This control will rule a non-specific effect of PTEN knockdown that interferes with membrane biology.

The authors should determine changes in RHOA activity and its potential role in the morphogenesis phenotype reported here because of the previously reported relationship between ARHGAP10 and RhoA.

The authors show PTEN loss decreases b-arrestin-ARHGAP10 interaction, however it is not clear how loss of PTEN affects the interaction between b-arrestin and ARHGAP10. The authors should provide a clear insight into the mechanistic basis for this observation.

Reviewer #3:

The manuscript "PTEN controls glandular morphogenesis through a juxtamembrane b-arrestin1/ARHGAP10 scaffolding complex" from Javadi et al., builds on previous molecular data demonstrating that (i) b-arr1 can interact with the C2 domain of PTEN to regulate phosphatase-independent PTEN functions, and (ii) that b-arr1 binds and suppresses the GAP ARHGAP10/21, to reveal that glandular morphogenesis is controlled through juxtamembrane b-arr1/ARHGAP10/CDC42 dynamics. The study adds new knowledge to how PTEN and b-arr1 can coordinately regulate cell polarity in multicellular architecture, and that changes in this complex drives morphological defects in relevant colorectal cancer models. While the work is well done, some critical controls are missing and some mechanistic points could be further developed and discussed to strengthen the findings.

In Figure 1 b-arr1 staining in the PTEN-deficient lines is weak. Enhanced contrast images should also be shown so the reader can more readily analyze the panels. Why is there a discrepancy between the blot data (in Figure 1 small effect on the b-arr1 signal) and the IF data (in Figure 1 large decrease in signal)? The authors also state in Figure 5 that the isolated C2 domain enhanced b-arr1 immunoreactivity and decreased ARHGAP10 immunoreactivity, but do not come back to it in the discussion. Please discuss these findings further.

In Figure 3 another gel should be shown for the ARHGAP10 blot in Caco-2 shPTEN cells, as there appears to be a technical issue that would affect quantification. How many times was this experiment performed? The same goes for Figure 4. Also, in all coIP experiments in Figure 3, control experiments should be performed with non-relevant control antibodies. Another issue, in Figure 3, is that blots should be shown to demonstrate equivalent expression of C2 and C2MCBR3. The same holds for Figure 4 and D.

What is the effect of CBR3 mutation in the FL PTEN in Figure 4?

The authors use a previously characterized cell-permeant disruptor peptide to inhibit the b-arr1/ARHGAP10 complex (Anthony et al., MCB 2011). It would also be interesting to test the effect of inhibiting the b-arr1/PTEN complex, which would additionally strengthen the molecular data. One way to do this would be to use previously characterized PTEN mutants that display differential binding to b-arr1 in a phosphatase-independent setting (Lima-Fernandes et al. EMBO J 2011). In a catalytically dead context (C124S), mutation of Thr 383 to Ala increases b-arr1/PTEN binding whereas mutation of Ser380, Thr382, Thr 383 and Ser385 altogether to Ala decreases b-arr1/PTEN binding. If the authors can recapitulate this difference in binding, in their colorectal model systems, the effect of reintroducing these mutants on active Cdc42 levels and morphology phenotypes could be compared.

The discussion should be expanded to give a more comprehensive overview on GPCR/b-arr/PTEN regulation and the functional effects previously attributed to the b-arr1/ARHGAP10 (stress fibre formation) and PTEN interactions (cell proliferation and migration). It would also be good to have a schematic of PTEN showing the C2 domain and the sequence of the mutant that was generated to create MCBR3. Similarly, a schematic of ARHGAP10, with the b-arr1 binding site and sequence of disruptor peptide used would be good to have.

Reviewer #4:

In this manuscript the authors investigate the role of the C2 domain of PTEN in controlling CDC42-dependent processes (i.e. spindle orientation and lumen formation) in Caco-2 3D cultures via regulation of β-Arrestin1/ARHGAP10 interaction. Because the PTEN C2 domain alone is sufficient to mediate the effect of PTEN the authors conclude that in this context PTEN functions as a scaffold rather than via its lipid modifying activity.

The non-catalytic function of PTEN via β-Arrestin1/ARHGAP10 is a novel an interesting result that is well supported by transfection (i.e. KD and rescue) experiments. There are however aspects of the mechanistic regulation of the proposed scaffolding function that are less convincing. Specifically, the co-IP experiments shown in Figure 3 and Figure 4 are not as convincing as the spindle orientation and lumen formation data. This may be due to technical limitations, co-IP/WB approaches are hardly quantitative and make detention of subtle differences difficult. It would be best if the authors could revisit their protein interaction experiments using other complementary methods such as proximity ligation, split systems or FRET.

[Editors’ note: what now follows is the decision letter after the authors submitted for further consideration.]

Thank you for submitting your article "*PTEN* controls glandular morphogenesis through a juxtamembrane *β-Arrestin1 /ARHGAP10* scaffolding complex" for consideration by *eLife*. Your article has been reviewed by two peer reviewers, and the evaluation has been overseen by a Reviewing Editor (Mohan Balasubramanian) and K VijayRaghavan as the Senior Editor. The following individual involved in review of your submission has agreed to reveal his identity: Michel Bagnat (Reviewer #2).

The reviewers have discussed the reviews with one another and the Reviewing Editor has drafted this decision to help you prepare a revised submission.

Summary:

The editors and the reviewers agreed that your work reporting the new functions of PTEN you describe vis-a-vis interactions with β-arrestin, ARHGAP21 interesting. However, a number of further points have been raised. We would like to invite you to revise the manuscript accordingly.

I would also like you to proofread the entire manuscript and figures carefully.

Essential revisions:

Point 2. Figure 1 and supplement. The image quality is still low, perhaps because of the pdf size, but some images also appear out of focus. They are not of adequate quality to show plasma membrane localization of β-arrestin1 or any change in its localization with and without PTEN expression or LPA stimulation. The higher magnification areas and use of β-arrestin1-GFP do not help here. Inclusion of a membrane marker is essential (e.g. GFP-K-Ras C-terminus) to indicate where the plasma membrane is. Although the authors rule out quantification, it is possible to draw line scans across the cells, and determine whether or not β-arrestin1 co-localizes with the plasma membrane marker. Confocal z-stacks might help to indicate the localization better. Why are the scale bars in a different place in each image? All images should be at the same scale, and then one scale bar in one image is adequate.

---

## [Author Response]

[Editors’ note: the author responses to the first round of peer review follow.]

*Reviewer #1:*

*In this paper, Javadi et al. describe their investigation of the function of the tumor suppressor PTEN in morphogenesis and spindle orientation in 3D multicellular organoids. They concluded that the loss of PTEN or β arrestin causes impairment in both these functions and that these defects are restored by activation of Cdc42 through suppressing its GTPase activation protein, ARHGAP10. They also suggest that PTEN controls the interaction between β arrestin and ARHGAP10 through its C2 domain, and that disruption of this interaction leads to the inhibition of Cdc42. The data for the 3D organoid formation and the regulation for spindle orientation are very interesting and convincing. However, this reviewer has several problems with the presentation and reproducibility of the data, which makes scientific interpretation difficult. I believe the authors should meticulously review their manuscript and remedy the problems in this otherwise potentially very interesting and important work. Specifically:*
*1) In Figure 1, the authors show that knockout or knockdown of PTEN leads to decreased membrane-bound β arrestin levels and increased membrane-bound ARHGAP10 levels, as evidenced by a crude membrane fractionation assay. However, the amounts of membrane-bound β arrestin are inconsistent between Figure 1 (showing a 50% decrease) and Figure 1 (showing almost no decrease). To compare each group shown in Figure 1, statistical analysis was performed only by ANOVA without multiple comparisons. Furthermore, for the same experiment shown in Figure 1—figure supplement 3, Western blotting does not show a decrease in membrane-bound β arrestin levels in PTEN KO, although the quantification shows a 20% decrease at the basal level (before LPA stimulation). Again, statistical analysis is incomplete.*

We thank the reviewer for his assessment of our manuscript and helpful criticisms. We have now repeated the membrane fractionation, immunoblot, Co-IP studies and added statistics. While we found experimental variation in absolute densitometry values, differences of juxtamembrane *β-Arrestin1* and *ARHGAP10* protein levels between *PTEN*-expressing and deficient cells are significant in the new experiments. Our findings are shown in new Figure 1, Figure 1—figure supplement 3 and new Figure 1—figure supplements 5-9. We have repeated statistical analysis of the new experimental findings and conducted ANOVA Tukey post hoc tests for Figure 1 and other figures. We show significant differences of juxtamembrane *β-Arrestin1* between *PTEN –* expressing and – deficient cells both in basal conditions and after LPA stimulation. We have supplemented the Western blots by membrane localization studies of fluorescently tagged *β-Arrestin1* constructs (Figure 1 and Figure 1—figure supplement 1). We have also supplemented Co-IPs by more sensitive proximity ligation assays to demonstrate protein complex formation in vivo (See below).

*2) The intracellular localization of β arrestin is shown in Figure 1 and Figure 1—figure supplement 1. The images are improperly modified; the size and cropped area are not the same in each image. Also, in each case, the image quality is very low and pixelated, making it difficult to evaluate their plasma membrane localization.*

In response to this important concern, we have now conducted new intracellular localization studies for Figure 1 using a GFP-labelled *βArrestin1* construct *vs* GFP control. We have also used high power inserts to better demonstrate differences in membrane localization of *β-Arrestin1*-GFP between *PTEN*-expressing and -deficient cells, after LPA stimulation. These studies are supported by new quantitative cell fractionation and immunoblot studies that show significant membrane expression differences. These additional experiments more clearly show effects of *PTEN* on membrane localization of *β-Arrestin1*.

*3) In Figure 1, the authors suggest that in the PTEN knockdown, the β arrestin level in the membrane fraction decreases while the ARHGAP10 level increases. They also show in Figure 4 that expression of the C2 domain of PTEN restores these changes caused by PTEN KD/KO. There is no ARHGAP10 in the cytosol fraction. The authors should include essential control to show that the total amounts of β arrestin and ARHGAP10 are not affected by PTEN KD/KO. What percentage of cytosolic and membrane proteins are loaded?*

To address these key points, we conducted expression assays of total *β-Arrestin1* and *ARHGAP10* in whole cell lysates in *PTEN*-expressing and -deficient cells, shown in new Figure 1—figure supplements 4-8. While we did find expression differences in total lysate, we found proportionately greater differences in juxtamembrane localization of *β-Arrestin1* and *ARHGAP10*, between *PTEN*- expressing and-deficient cells particularly after LPA stimulation. These findings indicate that *PTEN* synergizes with LPA in active membrane recruitment of *β-Arrestin1*. In transfection experiments of GFPtagged -C2 and -C2-MCBR3, we now include separate panels for GFP that show equivalent transfection efficiency and expression levels. Appropriate loading controls were used. We found very low concentrations of *ARHGAP10* with in the cytosolic fraction, in all of our experiments. We loaded equivalent membrane and cytosol fractions in immunoblots and CoIPs. These points have been added to the new manuscript.

*4) In a related note, the loss of PTEN may simply change the expression levels of β arrestin and ARHGAP10 and not their membrane localizations. This possibility is supported by many data (intensity of β arrestin in Figure 1, Western data for total β arrestin expression in cells expressing C2 in Figure 3, and increased intensity of β arrestin and decreased intensity of ARHGAP10 in C2-expressing 3D organoids in Figure 5).*

This important criticism has been partly addressed in our response to point 3. While expression levels differ between *PTEN*-expressing and -deficient total cell lysates, there are greater differences in membrane localization between *PTEN*-expressing and -deficient cells after LPA stimulation. Furthermore, the *PTEN* C2 domain CBR3 interfacial membranetargeting loop is essential for *PTEN/β-Arrestin1/ARHGAP10* scaffolding and 3D multicellular morphogenesis. *PTEN/β-Arrestin1* morphogeneic effects are mediated by suppression of *ARHGAP10* that is predominantly localized within the crude membrane fraction. Hence, the *PTEN* C2 domain membrane targeting function is crucial for *PTEN* noncatalytic regulation of morphogenenic processes. These points have been discussed in the new manuscript.

*5) In Figure 3, the authors show that loss of PTEN decreases β arrestin-ARHGAP10 interaction. Ectopic expression of the C2 domain, but not the MCBR3 mutant, rescues this phenotype. The expression levels of C2 and MCBR3 should be examined to exclude the possibility that they have different expression levels and that is the reason for their distinct rescuing activities. They should also test whether C2 or MCBR3 interacts with the β arrestin-ARHGAP10 complex.*

To address these key points, we have shown equivalent expression of GFP-labelled C2, C2-MCBR3 and control GFP in transfected cells (New Figure 3; Figure 3—figure supplement 4). Notwithstanding the equivalent expression levels, C2 transfection promoted greater *β-Arrestin1-ARHGAP10* binding than C2-MCBR3. We complemented our CoIP studies and showed that the *PTEN* C2 domain binds *β-Arrestin1* and promotes *β-Arrestin1/ARHGAP10* interaction in vivo, using sensitive PLA techniques. To do this we used catalytically inactive full length *PTEN* constructs proposed to have an exposed C2 domain (*PTEN* CS124-T383A) and *PTEN* CS124-A4) (7) but that display differential *β-Arrestin1*-binding capacities (1), as well as C2 and C2-MCBR3 (New Figure 3; Figure 3—figure supplements 7 and 8) [also see responses below]. We also addressed C2 domain binding to the *βArrestin1/ARHGAP10* complex, by PLA. The key concept is that target binding between two colour-labelled proteins or protein complexes can generate a single composite signal (2,3). We have shown strong colocalization of C2GFP (green) and the *β-Arrestin1:ARHGAP10* complex (red) indicated by a composite yellow signal. This signal was not generated by C2-MCBR3-GFP or control -GFP (Figure 3). These findings have been discussed in the light of the reviewer’s criticism, in the revised manuscript.

*6) In Figure 4 through F, the authors show that expression of C2 leads to increases in the amounts of membrane-bound β arrestin and decreases in membrane-bound ARHGAP10 levels in PTEN KD/KO cells. How much C2 and MCBR3 are associated with the membrane fraction? Again, the authors need to show the total expression levels of these proteins.*

To address these essential points, we have included GFP expression data to show relative quantitation of each GFP-tagged protein in cytosol and membrane fractions of *PTEN-*deficient cells (Figure 4). Furthermore, relative cytosol and membrane localization of C2-GFP and C2MCBR3-GFP are shown in organotypic cultures, in Figure 5 and E.

*7) The increase in β arrestin in the membrane fraction by the expression of C2 shown in Figure 4 is not observed in Figure 4 and I.*

To address this concern, we have now repeated the immunoblot and Co-IP studies in membrane fractions of *PTEN*-deficient cells in new Figure 4. Our repeat experiments in triplicate now show greater *β-Arrestin1* in the membrane fraction, after transfection of C2 and full length *PTEN* but not C2-MCBR3 or *PTEN*-MCBR3.

*Reviewer #2:*

*Previous studies have shown that b-arrestin interacts with ARHGAP10 and negatively regulates its GAP activity. Here the authors show that PTEN positively regulates membrane localization of b-arrestin. Consistent with previously reported findings the authors show that the C2 domain of PTEN mediates the interaction with b-arrestin. In addition, the author investigate the biological significance of this pathway using caco2 3D culture system and investigate changes in spindle orientation during cell division and its effect of morphogenesis. This is an interesting study that connects PTEN with ARHGAP10 pathways and adds more insight into the previously known relationship between PTEN and b-arrestin. However, as outlined below, additional insights into the relationship between PTEN and b-arresting/ARHGAP10 interaction would increase the impact of the study.*
*Figure 1 – the authors state loss of PTEN results in "greater b-arresting but lower ARHGAP10 levels in membrane fractions" but the data presented show a lower b-arrestin and greater ARHGAP10.*

We thank the reviewer for his positive and helpful comments. We apologise if our findings were not wholly clear but our interpretation is that membrane *β-Arrestin1* is higher in Caco-2 than in *PTEN*-deficient Caco-2 Sh*PTEN.* Because of space limitations, we have referred to *Caco-2 ShPTEN* cellsas Sh*PTEN* in immunoblots, which may have caused confusion. We have now provided clearer descriptions in the text and figure legends. As indicated in our response to reviewer 1, we have repeated the studies for Figure 1 where higher *β-Arrestin1* and lower *ARHGAP10* are shown in membrane fractions of parental *PTEN* – expressing cells.

*Figure 1: Can the authors show a control membrane protein that can translocate to membrane upon a ligand stimulation irrespective of PTEN levels. This control will rule a non-specific effect of PTEN knockdown that interferes with membrane biology.*

We thank the reviewer for this suggestion and have investigated membrane localization of *E-Cadherin*, mediated by exposure to 1,25(OH)_2_D_3_ (4). We show equivalent 1,25(OH)_2_D_3 –_ mediated membrane localization of *E-Cadherin* in Caco-2 and Caco-2 Sh*PTEN* cells. We include the localization images (Author response image 1). As we already have many figures within the manuscript, we refer to the finding in the main text, as data not shown.

**Author response image 1. respfig1:** To investigate effects of *PTEN* on ligand- mediated enhancement of a membrane protein (Reviewer 2, point 2, Figure 1), we tested 100nM 1,25 (OH)_2_ Vit D_3_*vs* vehicle only control against membrane recruitment of *E-Cadherin* (4) in Caco-2 and Caco-2 Sh*PTEN* cells.

*The authors should determine changes in RHOA activity and its potential role in the morphogenesis phenotype reported here because of the previously reported relationship between ARHGAP10 and RhoA.*

To address this relevant suggestion, we have now assessed relationships between *β-Arrestin1*, *ARHGAP10* and *RhoA*. We show that *RhoA*-GTP intensity relates directly to *β-Arrestin1* and inversely to *ARHGAP10* expression. Similarly, *PTEN* knockdown suppresses *β-Arrestin*, enhances *ARHGAP10* and suppresses *RhoA-GTP* (Author response image 2). Again, we refer to this finding in the new manuscript as data not shown. Hence, *PTEN* acts as an upstream regulator of both *RhoA* and *Cdc2* GTPases via *β-Arrestin1* and *ARHGAP10*. While understanding of the specific roles of *CDC42* and *RHOA,* their respective tissue-specific GEFS and GAPS (5) and the particular subcellular, cellular and multicellular phenotypes that they control during 3D morphogenesis could greatly alter our comprehension of disease states, we feel that such a mechanistic dissection is beyond the scope of this manuscript.

**Author response image 2. respfig2:** 2A To investigate effects of the *PTEN/β-Arrestin1/ARHGAP10* cascade on *RHOA* (Reviewer 2; point 3), we assessed *RHOA*-GTP in Caco-2 *vs* Caco2 Sh*PTEN (*Sh*PTEN)* cells (First column).

*The authors show PTEN loss decreases b-arrestin-ARHGAP10 interaction, however it is not clear how loss of PTEN affects the interaction between b-arrestin and ARHGAP10. The authors should provide a clear insight into the mechanistic basis for this observation.*

We provide the following rationale for this important issue in the Discussion:

*“PTEN* interacts via its C2domain with *β-Arrestin1*. […] Collectively, these findings indicate that *PTEN* through its intact C2 domain binds *β-Arrestin1* and promotes *β-Arrestin1/ARHGAP* interactions”.

*Reviewer #3:*

*The manuscript "PTEN controls glandular morphogenesis through a juxtamembrane b-arrestin1/ARHGAP10 scaffolding complex" from Javadi et al., builds on previous molecular data demonstrating that (i) b-arr1 can interact with the C2 domain of PTEN to regulate phosphatase-independent PTEN functions, and (ii) that b-arr1 binds and suppresses the GAP ARHGAP10/21, to reveal that glandular morphogenesis is controlled through juxtamembrane b-arr1/ARHGAP10/CDC42 dynamics. The study adds new knowledge to how PTEN and b-arr1 can coordinately regulate cell polarity in multicellular architecture, and that changes in this complex drives morphological defects in relevant colorectal cancer models. While the work is well done, some critical controls are missing and some mechanistic points could be further developed and discussed to strengthen the findings.*
*In Figure 1 b-arr1 staining in the PTEN-deficient lines is weak. Enhanced contrast images should also be shown so the reader can more readily analyze the panels. Why is there a discrepancy between the blot data (in Figure 1 small effect on the b-arr1 signal) and the IF data (in Figure 1 large decrease in signal)? The authors also state in Figure 5 that the isolated C2 domain enhanced b-arr1 immunoreactivity and decreased ARHGAP10 immunoreactivity, but do not come back to it in the discussion. Please discuss these findings further.*

We thank the reviewer for his positive comments and important criticisms. We have partly addressed these points in our response to reviewer 1. We have repeated the fractionation studies and immunoblots of Figure 1 and have conducted membrane localization studies using GFP-tagged *β-Arrestin1* that show *PTEN* effects more clearly. Densitometry differences of *β-Arrestin1* and *ARHGAP10* in membrane fractions of *PTEN*-expressing and deficient cells are statistically significant. There are still some quantitative differences in membrane localization between new Figure 1 and Figure 1 that use different experimental methods. These can be attributed at least in part to different timescales of the experimental techniques. Figure 1 assesses protein accumulation in membrane fractions after 24 hrs culture while Figure 1 assesses differences at 5 minutes after LPA treatment *vs* control. The point concerning Figure 5 has been partly addressed by our response to reviewer 2 and our PLA studies that investigated *PTEN/βArrestin1/ARHGAP i*nteractions. We have also addressed relevant mechanisms in the Discussion as follows;

*“*C2 domains are phospholipid and protein binding modules involved in recruitment of signalling molecules to specific membrane regions. […]Hence, an intact C2 domain CBR3 loopmay be required for *PTEN* N-terminal promotion of *β-Arrestin1ARHGAP10* binding”.

*In Figure 3 another gel should be shown for the ARHGAP10 blot in Caco-2 shPTEN cells, as there appears to be a technical issue that would affect quantification. How many times was this experiment performed? The same goes for Figure 4. Also, in all coIP experiments in Figure 3, control experiments should be performed with non-relevant control antibodies. Another issue, in Figure 3, is that blots should be shown to demonstrate equivalent expression of C2 and C2MCBR3. The same holds for Figure 4 and D.*

To address these key points, we have repeated the immunoblots and Co-IPs for Figure 3 and Figure 4 and new quantitation figures. All immunoblots and CoIPs were conducted at least in triplicate. We have included a non-relevant control protein (IgG) in Figure 3, as the reviewer has suggested. We have conducted PLA studies to assess interactions with greater sensitivity. In new Figure 3 and Figure 4, we have included immunoblots of GFP to indicate GFP-C2 and GFP-C2MCBR3 transfection efficiency and transgene expression. Please note that in the revised manuscript, some figures have been renamed and Figure 3 is now new Figure 3, Figure 3 is now new Figure 3—figure supplement 4.

*What is the effect of CBR3 mutation in the FL PTEN in Figure 4?*

To address this point, we have repeated the CoIP and quantitation studies of Figure 4 and Figure 4 to include full length *PTEN* with the CBR3 mutation (*PTEN*-MCBR3). In that revised figure, we show that transfection of C2 or full length *PTEN* induced higher *β-Arrestin1/ARHGAP* interactions than C2-MCBR3 and *PTEN*-MCBR3 than respectively. We found that *PTEN*-MCBR3 did enhance *β-Arrestin1:ARHGAP10* binding in PLA assays but differences were not significant in CoIps. This may be a reflection of the greater sensitivity of PLA assays or the greater number of PLA experimental results for analysis (n = 35 cells for PLA *vs* single immunoblot densitometry values in triplicate) or both factors. The finding has been discussed in the manuscript text.

*The authors use a previously characterized cell-permeant disruptor peptide to inhibit the b-arr1/ARHGAP10 complex (Anthony et al., MCB 2011). It would also be interesting to test the effect of inhibiting the b-arr1/PTEN complex, which would additionally strengthen the molecular data. One way to do this would be to use previously characterized PTEN mutants that display differential binding to b-arr1 in a phosphatase-independent setting (Lima-Fernandes et al. EMBO J 2011). In a catalytically dead context (C124S), mutation of Thr 383 to Ala increases b-arr1/PTEN binding whereas mutation of Ser380, Thr382, Thr 383 and Ser385 altogether to Ala decreases b-arr1/PTEN binding. If the authors can recapitulate this difference in binding, in their colorectal model systems, the effect of reintroducing these mutants on active Cdc42 levels and morphology phenotypes could be compared.*

To address this excellent suggestion, we have obtained the *PTEN* mutants to which the reviewer refers, from our collaborator Dr Mark Scott, Inserm, Paris. We have confirmed the differential *β-Arrestin1* binding to these mutants in PLA studies (Figure 3) and have assessed their effects on *Cdc42* activation, in transfection studies. Unfortunately, we cannot test effects of these mutants upon morphology of our *PTEN-*deficient 3D model. This is because our Caco-2 Sh*PTEN* model was generated by stable expression of *PTEN* ShRNA targeted against the N-terminal coding region. Hence, we cannot express *PTEN* C124S-T383A in our Caco-2 Sh*PTEN* cells. Conversely, transfection of wt Caco-2 cells with *PTEN* CS-A4 would not bind endogenous *β-Arrestin1* and could not competitively inhibit endogenous *PTEN/β-Arrestin1* binding. This omission represents the only reviewers’ criticism that we were unable to address and it is because of intrinsic limitations of our model. We have however shown that *PTEN* C2 binds and promotes membrane recruitment of *β-Arrestin1,* enhances *β-Arrestin1:ARHGAP10* binding, upregulates *CDC42-GTP* and rescues morphogenic processes including spindle orientation, AM alignment and 3D multicellular gland formation. We show that phosphatase-independent *PTEN* binding of *β-Arrestin1* an unmasked C2 domain enhances *CDC42* activation. Conversely, suppression of *β-Arrestin1:ARHGAP10* binding by the targeted peptide, inhibited *CDC42* activation and disrupted morphogenesis. Hence, the case for *PTEN* phosphatase-independent regulation of morphogenesis through *β-Arrestin1:ARHGAP10* interaction and *CDC42* activation is robust.

*The discussion should be expanded to give a more comprehensive overview on GPCR/b-arr/PTEN regulation and the functional effects previously attributed to the b-arr1/ARHGAP10 (stress fibre formation) and PTEN interactions (cell proliferation and migration).*

We have now expanded the Discussion to include these points as follows:

*“PTEN* is frequently downregulated in human colorectal and other cancers, even in the absence of genetic loss or mutation. […]By modulation of *PTEN-β-Arrestin* interactions, GPCRs may influence *β-Arrestin*dependent GTPase activation, cytoskeletal dynamics, directional cell migration and cancer susceptibility”

*It would also be good to have a schematic of PTEN showing the C2 domain and the sequence of the mutant that was generated to create MCBR3. Similarly, a schematic of ARHGAP10, with the b-arr1 binding site and sequence of disruptor peptide used would be good to have.*

We have previously published a schematic of PTEN showing the C2 domain as well as PTEN-MCBR3 and C2-CBR3 (15). However, we now include a revised schematic (Figure 4) that is not a direct copy of the previously published figure. We obtained the MCBR3 mutants from Dr Nicholas Leslie, in Dundee who mutated 5 lysines in the CBR3 to Ala (26) as described by Lee et al. (10). We have confirmed this sequence order and include a description of the mutant sequence. We also include a schematic of ARHGAP10, with the b-arr1 binding site and sequence of disruptor peptide used, as the reviewer has suggested (Figure 6).

*Reviewer #4:*

*In this manuscript the authors investigate the role of the C2 domain of PTEN in controlling CDC42-dependent processes (i.e. spindle orientation and lumen formation) in Caco-2 3D cultures via regulation of β-Arrestin1/ARHGAP10 interaction. Because the PTEN C2 domain alone is sufficient to mediate the effect of PTEN the authors conclude that in this context PTEN functions as a scaffold rather than via its lipid modifying activity.*
*The non-catalytic function of PTEN via β-Arrestin1/ARHGAP10 is a novel an interesting result that is well supported by transfection (i.e. KD and rescue) experiments. There are however aspects of the mechanistic regulation of the proposed scaffolding function that are less convincing. Specifically, the co-IP experiments shown in Figure 3 and Figure 4 are not as convincing as the spindle orientation and lumen formation data. This may be due to technical limitations, co-IP/WB approaches are hardly quantitative and make detention of subtle differences difficult. It would be best if the authors could revisit their protein interaction experiments using other complementary methods such as proximity ligation, split systems or FRET.*

We thank the reviewer for these excellent suggestions and have now conducted proximity ligation assays that show differential *β-Arrestin1* binding *to PTEN vs PTEN*-MCBR3 and to C2 *vs* C2-MCBR3. Furthermore we show differential effects of *PTEN vs PTEN*-MCBR3 and C2 *vs* C2-MCBR3 on *β-Arrestin1/ARHGAP10* interactions (Figure 3). To validate the proximity ligation data we included two *PTEN* mutant constructs (*PTEN* CS-T383A and *PTEN* CS-A4) that we obtained from our collaborator, Dr Mark Scott, Inserm, that respectively have or do not have of *β-Arrestin1* binding capacity. As indicated in the response to reviewer 3, bioluminescence resonance energy transfer (BRET) analyses of CS-T383A and PTEN CS-A4 show different conformations consistent with proteinbinding differences. These findings support the main finding of our manuscript that *PTEN* C2 domain controls threedimensional multicellular morphogenesis through juxtamembrane *β-Arrestin1/ARHGAP10/CDC42* scaffolding.

[Editors' note: the author responses to the re-review follow.]

*Summary:*

*The editors and the reviewer agreed that your work reporting the new functions of PTEN you describe vis-a-vis interactions with β-arrestin, ARHGAP21 interesting. However, a number of further points have been raised. We would like to invite you to revise the manuscript accordingly.*
*I would also like you to proofread the entire manuscript and figures carefully.*

We thank the editors and reviewers for their supportive comments and very helpful criticisms that have enabled substantive improvement of the manuscript. After careful proofreading, we have corrected all typing and other errors in the text, figure legends and figures. We address reviewers and editors specific comments with an itemised response, outlined below. In addition, we have also revised most figures to improve perception of detail. Alterations include new experiments and enlarged image sizes of β-Arrestin1 colocalization with a plasma membrane marker within new Figure 1. We also include line scans and 2 Z-stack reconstructions. In Figure 2 and Figure 5, we better show spindle alignment by use of solid and interrupted white arrows to show the angles of spindle orientation towards gland centres, in high power spindle views. In Figure 3 and Figure 7 we have enlarged panel sizes to better show detail in proximity ligation assays (PLA) and organoid studies respectively. In all of the blots we have included molecular weight indicators as recommended. We include total lysate expression of β-Arrestin1 and ARHGAP21 in blots as recommended. In quantitative graphs, we express β-Arrestin1 and ARHGAP21 protein levels in membrane fractions as a ratio of total expression values in cell lysates, as recommended. We clearly indicate this change in Y-axis graph labels. To aid clarity, we have repeatedly used the terms Caco-2 ShPTEN (ShPTEN), PTEN-/- HCT116 (PTEN -/-) and PTEN C2 (C2) to indicate the cellular or gene origins of abbreviated forms of ShPTEN, PTEN -/- and C2 respectively.To aid visibility, we have also placed just one 20 µm scale bar in one image of each composite figure, as recommended.

*Essential revisions:*

*Point 2. Figure 1 and supplement. The image quality is still low, perhaps because of the pdf size, but some images also appear out of focus. They are not of adequate quality to show plasma membrane localization of β-arrestin1 or any change in its localization with and without PTEN expression or LPA stimulation. The higher magnification areas and use of β-arrestin1-GFP do not help here. Inclusion of a membrane marker is essential (e.g. GFP-K-Ras C-terminus) to indicate where the plasma membrane is. Although the authors rule out quantification, it is possible to draw line scans across the cells, and determine whether or not β-arrestin1 co-localizes with the plasma membrane marker. Confocal z-stacks might help to indicate the localization better. Why are the scale bars in a different place in each image? All images should be at the same scale, and then one scale bar in one image is adequate.*

We have followed reviewers’ advice in points 2 and 3 for these essential revisions. We used wheat germ agglutinin (WGA) as a plasma membrane marker) and show colocalization of β-Arrestin1-mCherry with Alexa- 488-labelled WGA in PTEN-expressing or -deficient cells, with or without LPA stimulation. As recommended, we also use line scans and Z-stack reconstructions. We use mCherry only controls. To improve visibility, we have also placed just one 20 µm scale bar in each composite figure, as suggested.

References

1. Lima-Fernandes E, Enslen H, Camand E, Kotelevets L, Boularan C, Achour L, et al. Distinct functional outputs of PTEN signalling are controlled by dynamic association with β-arrestins. Embo J. [Research Support, Non-U.S. Gov't]. 2011 Jul 6;30(13):2557-68.

2. Agrawal A, Deo R, Wang GD, Wang MD, Nie S. Nanometer-scale mapping and single-molecule detection with color-coded nanoparticle probes. Proc Natl Acad Sci U S A. [Research Support, N.I.H., Extramural Research Support, Non-U.S. Gov't]. 2008 Mar 4;105(9):3298-303.

3. Huang B, Babcock H, Zhuang X. Breaking the diffraction barrier: superresolution imaging of cells. Cell. [Research Support, N.I.H., Extramural Research Support, Non-U.S. Gov't]. 2010 Dec 23;143(7):1047-58.

4. Palmer HG, Gonzalez-Sancho JM, Espada J, Berciano MT, Puig I, Baulida J, et al. Vitamin D(3) promotes the differentiation of colon carcinoma cells by the induction of E-cadherin and the inhibition of β-catenin signaling. J Cell Biol. 2001 Jul 23;154(2):369-87.

5. Cherfils J, Zeghouf M. Regulation of small GTPases by GEFs, GAPs, and GDIs. Physiological reviews. [Research Support, Non-U.S. Gov't Review]. 2013 Jan;93(1):269-309.

6. Rahdar M, Inoue T, Meyer T, Zhang J, Vazquez F, Devreotes PN. A phosphorylation-dependent intramolecular interaction regulates the membrane association and activity of the tumor suppressor PTEN. Proc Natl Acad Sci U S A. 2009;106(2):480-5.

7. Raftopoulou M. Regulation of Cell Migration by the C2 Domain of the Tumor Suppressor PTEN. Science. 2004;303(5661):1179-81.

8. Lima-Fernandes E, Misticone S, Boularan C, Paradis JS, Enslen H, Roux PP, et al. A biosensor to monitor dynamic regulation and function of tumour suppressor PTEN in living cells. Nat Commun. [Research Support, Non-U.S. Gov't]. 2014;5:4431.

9. Corbalan-Garcia S, Gomez-Fernandez JC. The C2 domains of classical and novel PKCs as versatile decoders of membrane signals. BioFactors (Oxford, England). 2010;36(1):1-7.

10. Lee JO, Yang H, Georgescu MM, Di Cristofano A, Maehama T, Shi Y, et al. Crystal structure of the PTEN tumor suppressor: implications for its phosphoinositide phosphatase activity and membrane association. Cell. 1999;99(3):323-34.

11. Raftopoulou M, Hall A. Cell migration: Rho GTPases lead the way. Dev Biol.

2004;265(1):23-32.

12. Galvez-Santisteban M, Rodriguez-Fraticelli AE, Bryant DM, Vergarajauregui S, Yasuda T, Banon-Rodriguez I, et al. Synaptotagmin-like proteins control the formation of a single apical membrane domain in epithelial cells. Nat Cell Biol. 2012 Jul 22;14(8):838-49.

13. McConnachie G, Pass I, Walker SM, Downes CP. Interfacial kinetic analysis of the tumour suppressor phosphatase, PTEN: evidence for activation by anionic phospholipids. Biochem J. 2003;371(Pt 3):947-55.

14. Leslie NR, Yang X, Downes CP, Weijer CJ. PtdIns(3,4,5)P(3)-dependent and -independent roles for PTEN in the control of cell migration. Curr Biol. 2007;17(2):115-25.

15. Jagan IC, Deevi RK, Fatehullah A, Topley R, Eves J, Stevenson M, et al. PTEN phosphatase-independent maintenance of glandular morphology in a predictive colorectal cancer model system. Neoplasia. [Research Support, Non-U.S. Gov't]. 2013 Nov;15(11):1218-30.

16. Walker SM, Leslie NR, Perera NM, Batty IH, Downes CP. The tumour-suppressor function of PTEN requires an N-terminal lipid-binding motif. Biochem J. 2004;379(Pt 2):301-7.

17. Naguib A, Cooke JC, Happerfield L, Kerr L, Gay LJ, Luben RN, et al. Alterations in PTEN and PIK3CA in colorectal cancers in the EPIC Norfolk study: associations with clinicopathological and dietary factors. BMC Cancer. 2011;11:123.

18. Salmena L, Carracedo A, Pandolfi PP. Tenets of PTEN tumor suppression. Cell. 2008;133(3):403-14.

19. Song P, Zhang M, Wang S, Xu J, Choi HC, Zou MH. Thromboxane A2 receptor activates a Rho-associated kinase/LKB1/PTEN pathway to attenuate endothelium insulin signaling. J Biol Chem. [Research Support, N.I.H., Extramural Research Support, Non-U.S. Gov't]. 2009 Jun 19;284(25):17120-8.

20. Song MS, Salmena L, Pandolfi PP. The functions and regulation of the PTEN tumour suppressor. Nat Rev Mol Cell Biol. 2012 May;13(5):283-96.

21. Shukla AK, Violin JD, Whalen EJ, Gesty-Palmer D, Shenoy SK, Lefkowitz RJ. Distinct conformational changes in β-arrestin report biased agonism at seven-transmembrane receptors. Proc Natl Acad Sci U S A. [Research Support, N.I.H.,

Extramural Research Support, Non-U.S. Gov't]. 2008 Jul 22;105(29):9988-93.

22. Urs NM, Jones KT, Salo PD, Severin JE, Trejo J, Radhakrishna H. A requirement for membrane cholesterol in the β-arrestin- and clathrin-dependent endocytosis of LPA1 lysophosphatidic acid receptors. J Cell Sci. [Research Support, N.I.H., Extramural]. 2005 Nov 15;118(Pt 22):5291-304.

23. Li TT, Alemayehu M, Aziziyeh AI, Pape C, Pampillo M, Postovit LM, et al. Β-arrestin/Ral signaling regulates lysophosphatidic acid-mediated migration and invasion of human breast tumor cells. Mol Cancer Res. [Research Support, Non-U.S. Gov't]. 2009 Jul;7(7):1064-77.

24. Decaillot FM, Kazmi MA, Lin Y, Ray-Saha S, Sakmar TP, Sachdev P. CXCR7/CXCR4 heterodimer constitutively recruits β-arrestin to enhance cell migration. J Biol Chem. [Research Support, N.I.H., Extramural Research Support, Non-U.S. Gov't]. 2011 Sep 16;286(37):32188-97.

25. Xiao K, McClatchy DB, Shukla AK, Zhao Y, Chen M, Shenoy SK, et al. Functional specialization of β-arrestin interactions revealed by proteomic analysis. Proc Natl Acad Sci U S A. [Research Support, N.I.H., Extramural Research Support, Non-U.S. Gov't]. 2007 Jul 17;104(29):12011-6.

26. Orchiston EA, Bennett D, Leslie NR, Clarke RG, Winward L, Downes CP, et al. PTEN M-CBR3, a versatile and selective regulator of inositol 1,3,4,5,6- pentakisphosphate (Ins(1,3,4,5,6)P5). Evidence for Ins(1,3,4,5,6)P5 as a proliferative signal. J Biol Chem. 2004;279(2):1116-22.

27. Georgescu MM, Kirsch KH, Kaloudis P, Yang H, Pavletich NP, Hanafusa H. Stabilization and productive positioning roles of the C2 domain of PTEN tumor suppressor. Cancer Res. 2000;60(24):7033-8.

28. Naguib A, Bencze G, Cho H, Zheng W, Tocilj A, Elkayam E, et al. PTEN functions by recruitment to cytoplasmic vesicles. Mol Cell. [Research Support, N.I.H., Extramural]. 2015 Apr 16;58(2):255-68.

29. Pertz O. Spatio-temporal Rho GTPase signaling – where are we now? J Cell Sci. 2010 Jun 1;123(Pt 11):1841-50.

30. Durgan J, Kaji N, Jin D, Hall A. Par6B and atypical PKC regulate mitotic spindle orientation during epithelial morphogenesis. J Biol Chem. 2011 Apr 8;286(14):12461-74.